# Massively multiplex single-molecule oligonucleosome footprinting

Nour J Abdulhay[1†], Colin P McNally[1†], Laura J Hsieh[1], Sivakanthan Kasinathan[2], Aidan Keith[1], Laurel S Estes[1], Mehran Karimzadeh[1,3], Jason G Underwood[4], Hani Goodarzi[1,5], Geeta J Narlikar[1], Vijay Ramani[1,5]*

[1]Department of Biochemistry & Biophysics, University of California San Francisco, San Francisco, United States; [2]Department of Pediatrics, Stanford University, Palo Alto, United States; [3]Vector Institute, Toronto, United States; [4]Pacific Biosciences of California Inc, Menlo Park, United States; [5]Bakar Computational Health Sciences Institute, San Francisco, United States

**Abstract** Our understanding of the beads-on-a-string arrangement of nucleosomes has been built largely on high-resolution sequence-agnostic imaging methods and sequence-resolved bulk biochemical techniques. To bridge the divide between these approaches, we present the single-molecule adenine methylated oligonucleosome sequencing assay (SAMOSA). SAMOSA is a high-throughput single-molecule sequencing method that combines adenine methyltransferase footprinting and single-molecule real-time DNA sequencing to natively and nondestructively measure nucleosome positions on individual chromatin fibres. SAMOSA data allows unbiased classification of single-molecular 'states' of nucleosome occupancy on individual chromatin fibres. We leverage this to estimate nucleosome regularity and spacing on single chromatin fibres genome-wide, at predicted transcription factor binding motifs, and across human epigenomic domains. Our analyses suggest that chromatin is comprised of both regular and irregular single-molecular oligonucleosome patterns that differ subtly in their relative abundance across epigenomic domains. This irregularity is particularly striking in constitutive heterochromatin, which has typically been viewed as a conformationally static entity. Our proof-of-concept study provides a powerful new methodology for studying nucleosome organization at a previously intractable resolution and offers up new avenues for modeling and visualizing higher order chromatin structure.

*For correspondence:
vijay.ramani@ucsf.edu

[†]These authors contributed equally to this work

## Introduction

The nucleosome is the atomic unit of chromatin. Nucleosomes passively and actively template the majority of nuclear interactions essential to life by determining target site access for transcription factors (*Spitz and Furlong, 2012*), bookmarking active and repressed chromosomal compartments via post-translational modifications (*Zhou et al., 2011*), and safeguarding the genome from mutational agents (*Papamichos-Chronakis and Peterson, 2013*). Our earliest views of the beads-on-a-string arrangement of chromatin derived from classical electron micrographs of chromatin fibres (*Olins and Olins, 1974*), which have since been followed by both light (*Huang et al., 2010*) and electron microscopy (*Ou et al., 2017*; *Song et al., 2014*) studies of in vitro-assembled and in vivo chromatin. In parallel, complementary biochemical methods using nucleolytic cleavage have successfully mapped the subunit architecture of chromatin structure at high resolution. These cleavage-based approaches can be stratified into those that focus primarily on chromatin accessibility (*Klemm et al., 2019*) (i.e. measuring 'competent' active chromatin [*Weintraub and Groudine, 1976*]), and those that survey nucleosomal structure uniformly across active and inactive genomic compartments. Understanding links between chromatin and gene regulation requires sensitive methods in all three

of these broad categories: in this study, we advance our capabilities in the third, focusing on a novel method to map oligonucleosomal structures genome-wide.

Nucleolytic methods for studying nucleosome positioning have historically used cleavage reagents (e.g. dimethyl sulphate [*Becker et al., 1986*], hydroxyl radicals [*Tullius, 1988*], nucleases [*Hewish and Burgoyne, 1973*]) followed by gel electrophoresis and/or Southern blotting to map the abundance, accessibility, and nucleosome repeat lengths (NRLs) of chromatin fibres (*Richard-Foy and Hager, 1987*). More recently, these methods have been coupled to high-throughput short-read sequencing (*Zentner and Henikoff, 2014*), enabling genome-wide measurement of average nucleosome positions. While powerful, all these methods share key limitations: measurement of individual protein-DNA interactions inherently requires destruction of the chromatin fibre and averaging of signal across many short molecules. These limitations extend even to single-molecule methyltransferase-based approaches (*Kelly et al., 2012*; *Krebs et al., 2017*; *Nabilsi et al., 2014*), which have their own biases (e.g. CpG/GpC bias; presence of endogenous $m^5dC$ in mammals; DNA damage due to bisulphite conversion), and are still subject to the short-length biases of Illumina sequencers. While single-cell (*Lai et al., 2018*; *Pott, 2017*) and long-read single-molecule (*Baldi et al., 2018*) genomic approaches have captured some of this lost contextual information, single-cell data are generally sparse and single-molecule Array-seq data must be averaged over multiple molecules. Ultimately, these limitations have hindered our understanding of how combinations of 'oligonucleosomal patterns' (i.e. discrete states of nucleosome positioning and regularity on single DNA molecules) give rise to active and silent chromosomal domains.

The advent of third-generation (i.e. high-throughput, long-read) sequencing offers a potential solution to many of these issues (*Shema et al., 2019*). Here, we demonstrate *S*ingle-molecule *A*denine *M*ethylated *O*ligonucleosome *S*equencing *A*ssay (SAMOSA), a method that combines adenine methyltransferase footprinting of nucleosomes with base modification detection on the PacBio single-molecule real-time sequencer (*Flusberg et al., 2010*) to measure nucleosome positions on single chromatin templates. We first present proof-of-concept of SAMOSA using gold-standard in vitro assembled chromatin fibres, demonstrating that our approach captures single-molecule nucleosome positioning at high-resolution. We next apply SAMOSA to oligonucleosomes derived from K562 cells to profile single-molecule nucleosome positioning genome-wide. Our data enables unbiased classification of oligonucleosomal patterns across both euchromatic and heterochromatic domains. These patterns are influenced by multiple epigenomic phenomena, including the presence of predicted transcription factor binding motifs and post-translational histone modifications. Consistent with estimates from previous studies, our approach reveals enrichment for long, regular chromatin arrays in actively elongating chromatin, and highly accessible, disordered arrays at active promoters and enhancers. Surprisingly, we also observe a large amount of heterogeneity within constitutive heterochromatin domains, with both mappable H3K9me3-decorated regions and human major satellite sequences harboring a mixture of irregular and short-repeat-length oliognucleosome types. Our study provides a proof-of-concept framework for studying chromatin at single-molecule resolution while suggesting a highly dynamic nucleosome-DNA interface across chromatin sub-compartments.

## Results

### Single-molecule real-time sequencing of adenine-methylated chromatin captures nucleosome footprints

Existing methyltransferase accessibility assays either rely on bisulfite conversion (*Kelly et al., 2012*; *Krebs et al., 2017*; *Nabilsi et al., 2014*) or use the Oxford Nanopore platform to detect DNA modifications (*Oberbeckmann et al., 2019*; *Shipony et al., 2020*; *Wang et al., 2019*). We hypothesized that high-accuracy PacBio single-molecule real-time sequencing could detect $m^6dA$ deposited on chromatin templates to natively measure nucleosome positioning. To test this hypothesis, we used the nonspecific adenine methyltransferase EcoGII (*Murray et al., 2018*) to footprint nonanucleosomal chromatin arrays generated through salt-gradient dialysis (*Figure 1—figure supplement 1*), using template DNA containing nine tandem repetitive copies of the Widom 601 nucleosome positioning sequence (*Lowary and Widom, 1998*) separated by ~46 basepairs (bp) of linker sequence followed by ~450 bp of sequence without any known intrinsic affinity for nucleosomes. After purifying DNA, polishing resulting ends, and ligating on barcoded SMRTBell adaptors, we subjected

libraries to sequencing on PacBio Sequel or Sequel II flow cells, using unmethylated DNA and methylated naked DNA as controls (*Figure 1A*). After filtering low quality reads, we analyzed a total of 33,594 single molecules across all three conditions. Across both platforms, we observed higher average interpulse duration (IPD) in samples exposed to methyltransferase, consistent with a rolling circle polymerase 'pausing' at methylated adenine residues in template DNA (*Figure 1—figure supplement 2*). Further inspection of footprinted chromatin samples sequenced on either platform revealed strong specificity for altered IPD values only at thymines falling outside Widom 601 repeat sequences, in contrast with fully methylated naked template and unmethylated controls (*Figure 1—figure supplement 3A,B*). These patterns were subtly influenced by the associated 10-mer context of sequenced bases, consistent with possible enzymatic biases, but also previous observations of sequence-influenced shifts in polymerase kinetics (*Figure 1—figure supplement 4*; *Feng et al., 2013*). These results suggest that the PacBio platform can natively detect ectopic m⁶dA added to chromatinized templates.

We next developed a computational approach to assign a posterior probability describing the likelihood that an A/T basepair is methylated given IPD signals found within the same molecule (i.e. 'modification probability'). We then paired this approach with a simple peak-calling strategy to approximate nucleosomal dyad positions. To benchmark this pipeline, we first calculated the distance between called nucleosome dyads and expected 601 dyad positions (*Figure 1B*). Observed dyads were highly concordant with expected positions (median ±median absolute deviation [MAD] =4 ± 2.97 bp), consistent with our data accurately capturing the expected 601 dyad. We next

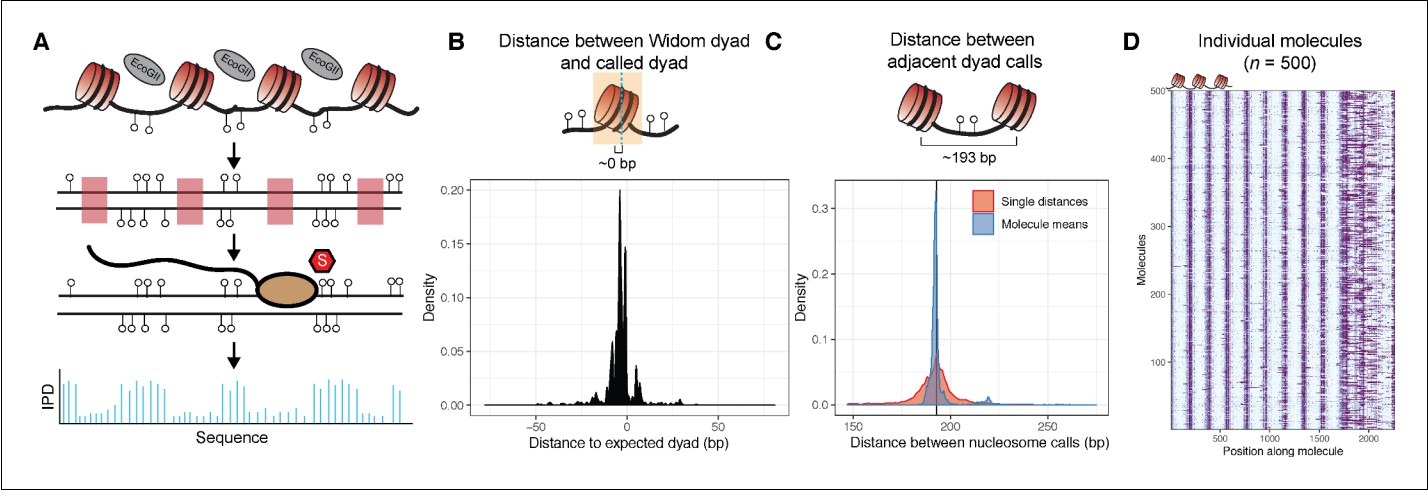

**Figure 1.** Overview of the single-molecule adenine methylated oligonucleosome sequencing assay (SAMOSA). (**A**) In the SAMOSA assay, chromatin is methylated using the nonspecific EcoGII methyltransferase, DNA is purified, and then subjected to sequencing on the PacBio platform. Modified adenine residues are natively detected during SMRT sequencing due to polymerase pausing, leading to an altered interpulse duration at modified residues. (**B**) SAMOSA data can be used to accurately infer nucleosome dyad positions given a strong positioning sequence. Shown are the distributions of called dyad positions with respect to the known Widom 601 dyad. Called dyads fall within a few nucleotides of the expected dyad position (median ±median absolute deviation [MAD]=4 ± 2.97 bp). (**C**) SAMOSA data accurately recapitulates the known nucleosome repeat lengths (NRL) of in vitro assembled chromatin fibres. Called NRLs are strongly concordant with the expected 193 repeat length (pairwise distance between adjacent dyads median ±MAD = 193±7.40 bp; single-molecule averaged repeat length median ±MAD = 192±1.30 bp). (**D**) Expected nucleosome footprints in SAMOSA data can be visually detected with single-molecule resolution (n = 500 sampled footprinted chromatin molecules).

The online version of this article includes the following figure supplement(s) for figure 1:

**Figure supplement 1.** Quality control of in vitro nucleosome arrays assembled through salt-gradient dialysis.

**Figure supplement 2.** Mean raw and quantile normalized interpulse durations for in vitro SAMOSA experiments.

**Figure supplement 3.** Adenine methylation by the EcoGII enzyme is specific to accessible adenines and is protected against by the nucleosome.

**Figure supplement 4.** k-mer analysis of negative and positive control sequences reveals sequence biases of IPD measurements of EcoGII modified DNA.

**Figure supplement 5.** Average linker methylation and individually called dyad positions are qualitatively similar across the length of the nonnucleosomal array molecule.

**Figure supplement 6.** Unmethylated and fully methylated array DNA does not display the same periodic patterning of modified bases seen in methylated chromatin.

calculated the expected distances between nucleosomes given our dyad callset (i.e. a computationally defined nucleosome repeat length [NRL]; *Figure 1C*). Compared with the expected repeat length of 193 bp, our calculated results were similarly accurate at both two-dyad resolution (pairwise distance between adjacent dyads; median ±MAD = 193±7.40 bp) and averaged single-molecule resolution (median ±MAD = 192±1.30 bp). Both these measurements were qualitatively uniform across all molecules, independent of the positions of individual nucleosomes along individual array molecules (*Figure 1—figure supplement 5*). Finally, we directly visualized the modification probabilities of individual sequenced chromatin molecules and observed that modification patterns occurred in expected linker sequences (*Figure 1D*), and not in unmethylated or fully methylated control samples (*Figure 1—figure supplement 6A,B*). These results demonstrate that EcoGII footprinting is specific for unprotected DNA and that kinetic deviations observed in the data are not simply the result of primary sequence biases in the template itself. We hereafter refer to this approach as SAMOSA.

## SAMOSA captures regular nucleosome-DNA interactions in vivo through nuclease-cleavage and adenine-methylation simultaneously

Having shown that SAMOSA can footprint in vitro assembled chromatin fibres, we sought to apply our approach to oligonucleosomal fragments from living cells. Multiple prior studies have suggested that a light micrococcal nuclease (MNase) digest followed by disruption of the nuclear envelope and overnight dialysis can be used to gently liberate oligonucleosomes into solution without dramatically perturbing nucleosomal structure (*Ehrensberger et al., 2015*; *Gilbert and Allan, 2001*; *Gilbert et al., 2004*). After lightly digesting and solubilizing oligonucleosomes from human K562 nuclei, we methylated chromatin with EcoGII and sequenced methylated molecules on the Sequel II platform ($n$ = 1,855,316 molecules total; *Figure 2A*). As controls, we also shallowly sequenced deproteinated K562 oligonucleosomal DNA, and deproteinated oligonucleosomal DNA methylated with the EcoGII enzyme.

In vivo SAMOSA has several advantages compared to existing MNase- or methyltransferase-based genomic approaches. Our approach combines MNase-derived cuts flanking each fragment with methyltransferase footprinting of nucleosomes. MNase cuts mark the boundary of genomic 'barrier' elements like nucleosomes and can be tuned by modifying digestion conditions; accordingly, fragment length distributions from in vivo SAMOSA data display patterns emblematic of bulk nucleosomal array regularity (*Figure 2B*; *Figure 2—figure supplement 1*). Modification patterns of sequenced molecules then capture nucleosome-positioning information at single-molecule resolution; this is evident in single-molecule averages of modification probability in chromatin samples with respect to fully methylated and unmethylated controls (*Figure 2C*). While previous approaches for studying nucleosome regularity may capture each of the former information types, this method is, to our knowledge, the first that simultaneously captures the positioning of protein-DNA interactions through nucleolytic cleavage, and (through DNA methylation) the positioning of proximal protein-DNA interactions on the same single-molecule.

## SAMOSA enables unbiased classification of chromatin fibres on the basis of regularity and nucleosome repeat length

The relative abundance and diversity of oligonucleosome patterns across the human genome remains unknown. Given the single-molecule nature of SAMOSA, we speculated that our data could be paired with a state-of-the-art community detection algorithm to systematically cluster footprinted molecules on the basis of single-molecule nucleosome regularity and NRL (i.e. 'oligonucleosome patterns'). To ease detection of signal regularity on single molecules, we computed autocorrelograms for each molecule in our dataset ≥500 bp in length, and subjected resulting values to unsupervised Leiden clustering (*Traag et al., 2019*). Cluster sizes varied considerably, but were consistent across both replicates, with each cluster containing 6.54% (Cluster 4)–20.1% (Cluster 1) of all molecules (*Figure 3A*). The resulting seven clusters (*Figure 3—figure supplement 1A*) capture the spectrum of oligonucleosome patterning genome-wide, stratifying the genome by both NRL and array regularity. Accounting for the coverage biases presented above, the measurements shown in *Figure 3A* provide a rough estimate of the equilibrium composition of the genome with respect to these patterns.

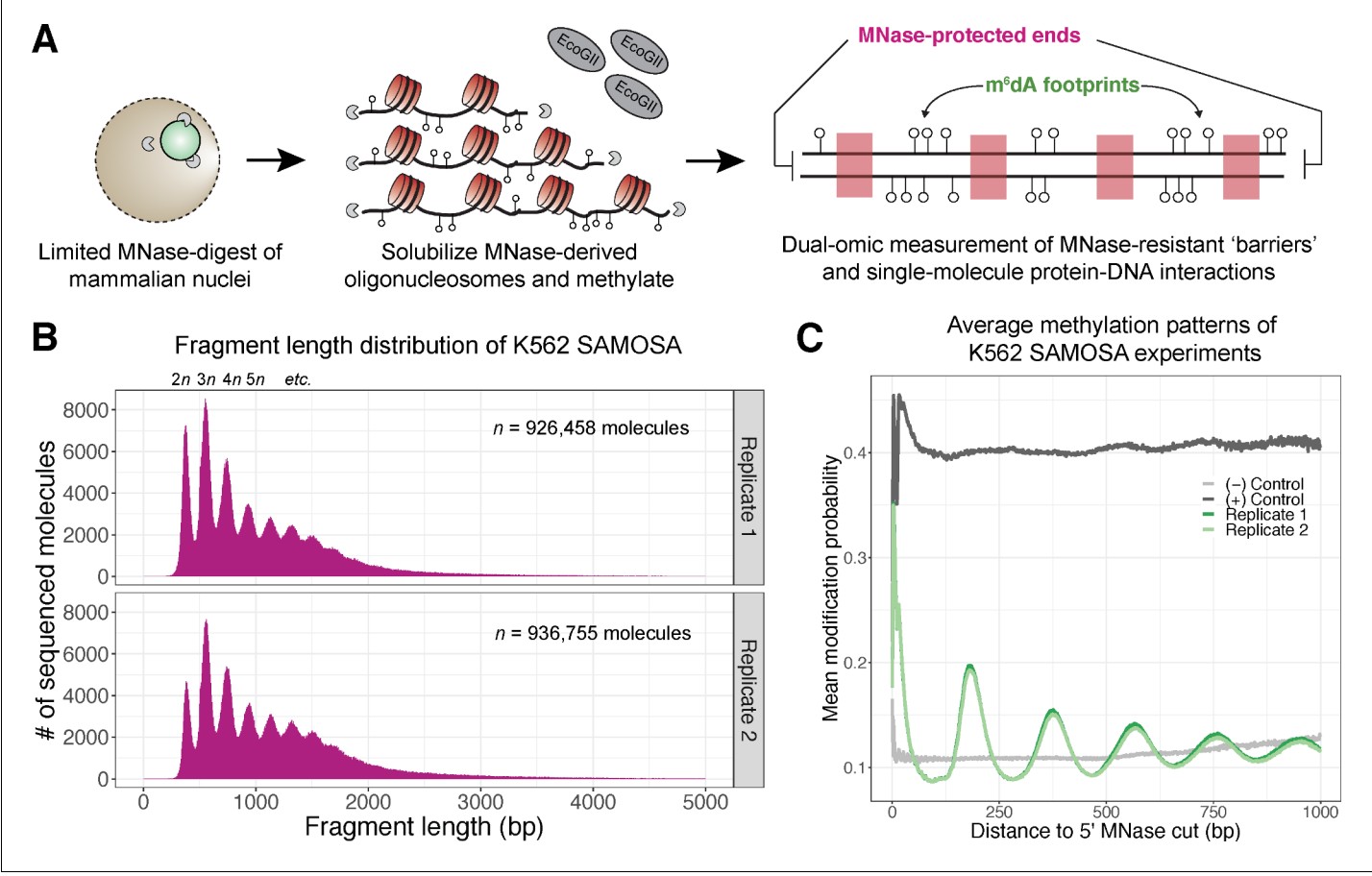

**Figure 2.** In vivo SAMOSA captures oligonucleosome structure by combining MNase digestion of chromatin with adenine methylation footprinting. (**A**) An overview of the in vivo SAMOSA protocol: oligonucleosomes are gently solubilized from nuclei using micrococcal nuclease and fusogenic lipid treatment. Resulting oligonucleosomes are footprinted using the EcoGII enzyme and sequencing on the PacBio platform. Each sequencing molecules captures two orthogonal biological signals: MNase cuts that capture 'barrier' protein-DNA interactions, and m6dA methylation protein-DNA footprints. (**B**) Fragment length distributions for in vivo SAMOSA data reveal expected oligonucleosomal laddering (bin size = 5 bp). (**C**) Averaged modification probabilities from SAMOSA experiments demonstrate the ability to mark nucleosome-DNA interactions directly via methylation. Modification patterns seen in the chromatin sample are not seen in unmethylated oligonucleosomal DNA or fully methylated K562 oligonucleosomal DNA.

The online version of this article includes the following figure supplement(s) for figure 2:

**Figure supplement 1.** Three additional K562 SAMOSA experimental conditions demonstrate the reproducibility of the technique for footprinting nucleosomes, and demonstrate the ability to tune SAMOSA fragment length distributions by altering MNase digestion conditions.

The diversity in nucleosome regularity and repeat length across these clusters is visually apparent when inspecting average modification probabilities of the 5' 1000 bp of each cluster (**Figure 3B**). To better annotate each of these clusters, we characterized each with respect to methylation extent and distribution of computed single-molecule NRLs. We first inspected the average modification probabilities of each molecule across clusters, finding that these averages were largely invariant (**Figure 3—figure supplement 1B**). This suggests that our clustering approach does not simply classify oligonucleosomes based on the amount of methylation on each molecule. We next estimated within-cluster heterogeneity in single-molecule NRLs using a simple peak-calling approach. We scanned each autocorrelogram for secondary peaks, and annotated the location of each peak to compute an estimated NRL. We then visualized these distributions as violin plots for each cluster (**Figure 3C**). Our data broadly fall into two categories: irregular clusters made up of molecules spanning multiple NRLs and lacking a strong regular periodicity, and highly regular clusters with defined single-molecule NRLs ranging from ~172 bp (i.e. chromatosome plus 5 bp DNA) to >200 bp. Based on the median NRLs and regularities inferred from these analyses, we named these clusters irregular-short (IRS), irregular-long (IRL), irregular-170 (IR170), regular repeat length 172 (NRL172), regular

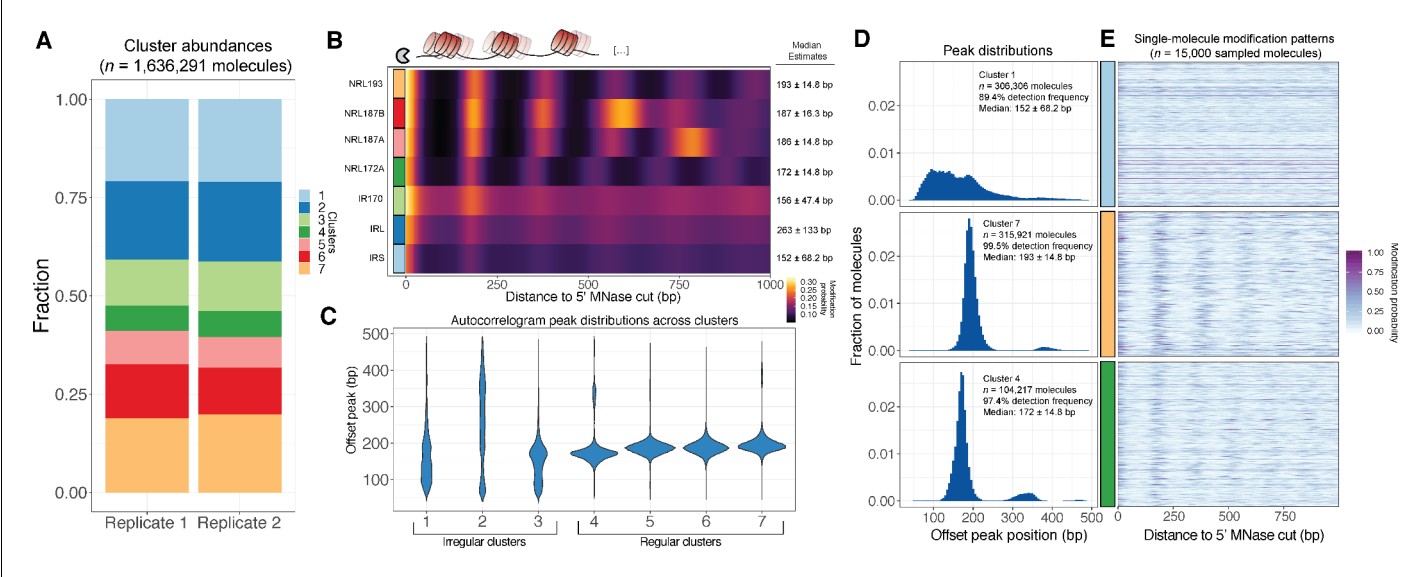

**Figure 3.** SAMOSA reveals distribution of oligonucleosome patterns genome-wide. (**A**) Stacked bar chart representation of the contribution of each cluster to overall signal across two replicate experiments in K562 cells. (**B**) Average modification probability as a function of sequence for each of the seven defined clusters. Left: Manually annotated cluster names based on NRL estimates computed by calling peaks on single-molecule autocorrelograms; Right: Median and median absolute deviation for single-molecule NRL estimates determined for each cluster. (**C**) Violin plot representation of the distributions of single-molecule NRL estimates for each cluster. Clusters can be separated into three 'irregular' and four 'regular' groups of oligonucleosomes. (**D**) Histogram of single-molecule NRL estimates for Clusters 1, 4, and 7, along with (**E**) 5000 randomly sampled molecules from each cluster.

The online version of this article includes the following figure supplement(s) for figure 3:

**Figure supplement 1.** Further characterization of clustered footprinted molecules.

repeat length 187A and B (NRL187A/B), and regular repeat length 192 (NRL192). The difference between irregular and regular clusters is clear when closely inspecting histograms of NRL calls from selected clusters (*Figure 3D*; *Figure 3—figure supplement 1C*), as well as the modification patterns on individual molecules (*Figure 3E*). Our analyses also varied with respect to the fraction of molecules per cluster where a secondary peak could be detected (0.50%–38.2% of molecules across specific clusters; *Figure 3—figure supplement 1D*). Failure to detect a peak within a single-molecule autocorrelogram could be due to multiple factors, including technical biases (e.g. random undermethylated molecules). We observed, however, that more 'missing' NRL estimates occurred in irregular clusters, suggesting that at least a fraction of failed peak calls occurred due to lack of intrinsic regularity along individual footprinted molecules. These analyses together demonstrate that SAMOSA data can be clustered in an unbiased manner, thus enabling estimates of the equilibrium composition of the genome with respect to oligonucleosome regularity and repeat length.

## SAMOSA captures the transient nucleosome occupancy of transcription-factor-binding motifs

We next explored the extent to which our data captures chromatin structure at predicted K562 transcription factor (TF)-binding sites (*ENCODE Project Consortium, 2012*). Both endo- and exo-nucleolytic MNase cleavage activities are obstructed by genomic protein-DNA contacts; resulting fragment-ends thus capture both nucleosomal- and TF-DNA interactions (*Henikoff et al., 2011*; *Ramani et al., 2019*). Inspection of cleavage patterns about six different TF-binding sites (CTCF, NRF1, NRSF/REST, PU.1, c-MYC, GATA1) (*Figure 4A–F*) revealed signal resembling traditional MNase-seq data, with fragment ends accumulating immediately proximal to predicted TF-binding motifs, and, in the case of some TFs (i.e. CTCF, REST, PU.1), showed characteristic patterns of phased nucleosomes. Further analysis of m6dA signal in sequenced molecules harboring motifs with at least 500 nucleotides of flanking DNA revealed examples of methyltransferase accessibility coincident with TF motifs (e.g. CTCF, NRF1, c-MYC), but also cases where single-molecule averages

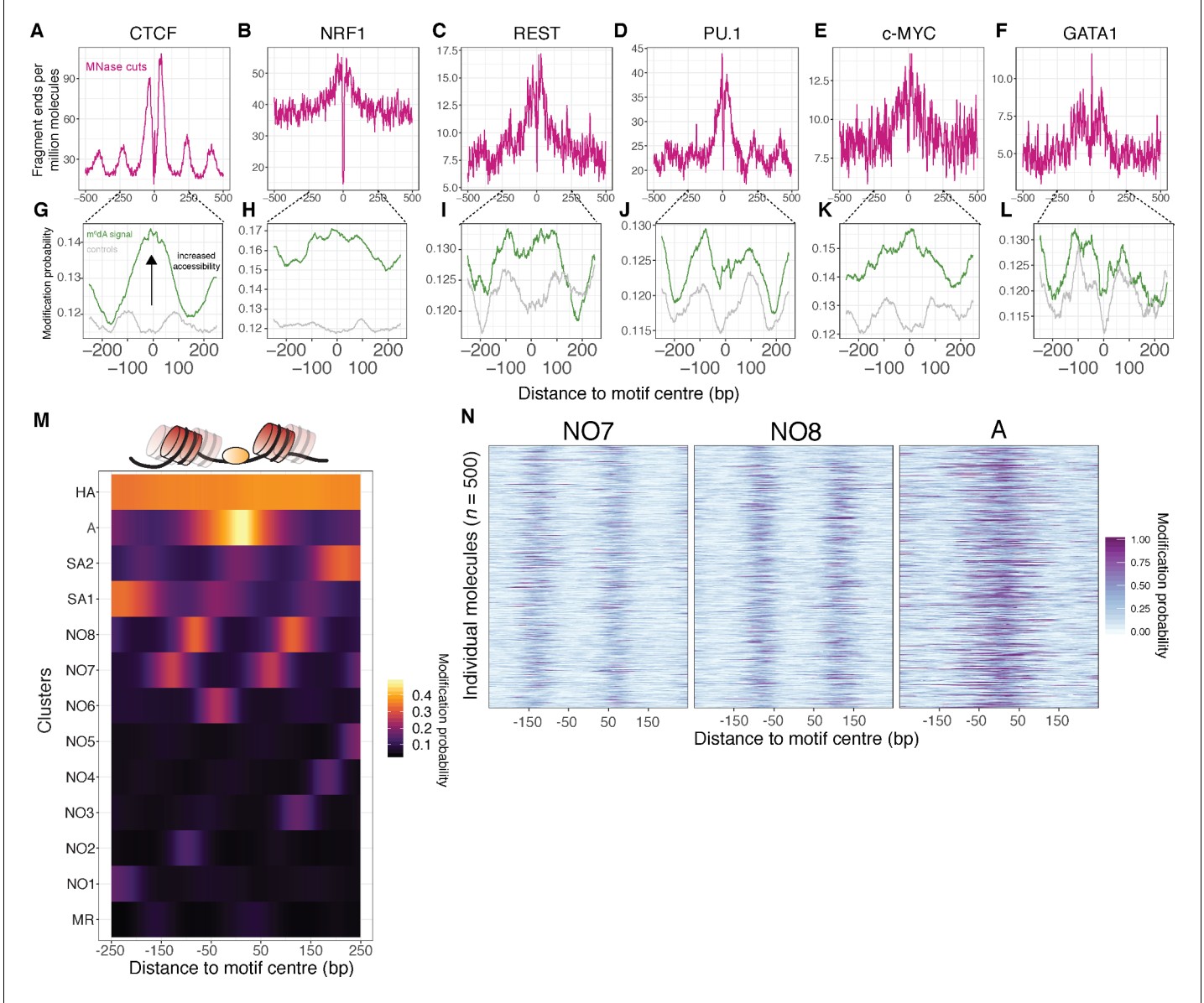

**Figure 4.** SAMOSA captures bulk and single-molecule evidence of transcription factor-DNA interaction simultaneously via two orthogonal molecular signals. (A-F) SAMOSA MNase-cut signal averaged over predicted CTCF, NRF1, REST, PU.1, c-MYC, and GATA1-binding motifs in the K562 epigenome. All binding sites were predicted from ENCODE ChIP-seq data. (G–L) $m^6dA$ signal for the same transcription factors, averaged over molecules containing predicted binding sites and at least 250 bases flanking DNA on either side of the predicted motif. Methylation patterns at predicted sites were compared against average profiles taken from randomly drawn molecules from GC%- and repeat-content-matched regions of the genome (calculated for each ENCODE ChIP-seq peak set). (M) Results of clustering motif-containing molecules using the Leiden community detection algorithm. Clusters were manually annotated as containing molecules that were: 'methylation resistant' (MR), nucleosome occupied (NO1-8), stochastically accessible (SA1-2), accessible (A), or hyper-accessible (HA). (N) Heatmap representation of single-molecule accessibility profiles for clusters NO7, NO8, and A (500 randomly sampled molecules per cluster).

The online version of this article includes the following figure supplement(s) for figure 4:

**Figure supplement 1.** Cluster sizes and numbers of motif-containing molecules for each transcription factor chosen for study.

demonstrated weak or no differential signal when compared to equal numbers of molecules drawn from random genomic regions matched for GC-percentage and repeat content (e.g. GATA1; *Figure 4G–L*). Importantly, our methylation data do not appear to capture TF 'footprints' as seen in DNase I, hydroxyl radical, or MNase cleavage data—this could be due to turnover of transcription

factors during our solubilization process, or owed to sterics, as EcoGII is roughly twice the molecular weight of *S. aureus* microccocal nuclease (*Murray et al., 2018*).

In theory, single-molecule footprinting data should distinguish nucleosome-bound and nucleosome-free states for molecules containing TF-binding sites. These accessibility patterns should be specific to TF-binding motifs (i.e. not present in control molecules matched for GC/repeat content). To test whether our assay captured such signal, we clustered all molecules shown in *Figure 4G–L* (including control molecules) using Leiden clustering, using modification probabilities extracted in a 500 bp window surrounding the predicted motif site/control site. In total, we defined 13 discrete states of template accessibility across all surveyed molecules (*Figure 4M*; cluster sizes shown in *Figure 4—figure supplement 1*). We interpreted these states on the basis of methyltransferase accessibility as: methyltransferase-resistant motifs (MR); nucleosome-occluded motifs (NO1-8); stochastically accessible motifs (wherein motif accessibility is slightly elevated near the DNA entry/exit point of a footprinted nucleosome; SA1-2); accessible motifs (A); and hyper-accessible motifs (HA). Notably, the patterns within these clusters were evident at single-molecule resolution (*Figure 4N*). Most transcription factors (excepting PU.1 and GATA1—the latter of which may productively bind nucleosomal DNA [*Zaret and Carroll, 2011*]) were significantly enriched for specific states as defined above, and all control regions were markedly depleted for molecules harboring the accessible 'A' and 'HA' states, hinting at the biological relevance of these patterns (*Figure 5A*). We speculate that the broad distribution of these states across both TF-binding sites and controls represent distributions of nucleosome 'registers' surrounding typical transcription factor binding motifs (i.e. states MR; NO-1–8). A fraction of these registers (i.e. states SA1/2) may stochastically permit transcription factor binding (perhaps through transient unwrapping of the nucleosome [*Polach and Widom, 1995*]), enabling formation of a new nucleosome register (i.e. state 'A'), and subsequent generation of a highly accessible state ('HA'; model illustrated in *Figure 5B*). The relative fraction of molecules in an 'SA' state could conceivably be modulated by TF intrinsic properties (e.g. ability to bind partially nucleosome-wrapped DNA [*Zaret and Mango, 2016*]), or extrinsic factors (e.g. local concentration of ATP-dependent chromatin remodeling enzymes [*Narlikar et al., 2013*]). While correlation of our replicates demonstrates the reproducibility and robustness of these findings (*Figure 5—figure supplement 1*), future experimental follow-up coupling our protocol with perturbed biological systems and deeper sequencing are necessary to quantitatively interrogate this model.

## Heterogeneous oligonucleosome patterns comprise human epigenomic domains

Short-read and long-read sequencing of nucleolytic fragments in mammals have suggested that NRLs vary across epigenomic domains, with euchromatin harboring shorter NRLs on average and heterochromatic domains harboring longer NRLs (*Gaffney et al., 2012*; *Snyder et al., 2016*; *Valouev et al., 2011*), but the relative heterogeneity of these domains remains unknown. We speculated that SAMOSA data could be used to estimate single-molecule oligonucleosome pattern heterogeneity across epigenomic domains. We revisited the seven oligonucleosome patterns defined above, and examined the distribution of patterns across collections of single molecules falling within ENCODE-defined H3K4me3, H3K4me1, H3K36me3, H3K27me3, and H3K9me3-decorated chromatin domains. To control for the impact of GC-content on these analyses, we also included GC-/repeat content matched control molecules for each epigenomic mark surveyed. Furthermore, to take advantage of the long-read and relatively unbiased nature of our data, we also incorporated molecules deriving from typically unmappable human alpha, beta, and gamma satellite DNA sampled directly from raw CCS reads.

We visualized the relative heterogeneity of these domains and controls in two ways: using histograms of computed single-molecule NRL estimates (*Figure 6A*), and by using stacked bar graphs to visualize cluster membership (*Figure 6B*). A striking finding of our analyses was that each epigenomic domain surveyed was comprised of a highly heterogeneous mixture of oligonucleosome patterns. In most cases, these patterns differed only subtly from control molecules with respect to regularity and NRL. In specific cases, we observed small effect shifts in the estimated median NRLs for specific domains—for example, a shift of ~5 bp (180 bp vs. 185 bp) in H3K9me3 chromatin with respect to random molecules, and a shift of ~4 bp (182 bp vs 186 bp) for H3K36me3. These shifts were also evident in the fraction of molecules with successful peak calls: H3K4me3 decorated chromatin, for example, had markedly fewer (78.0% vs 88.6%) successful calls compared to control

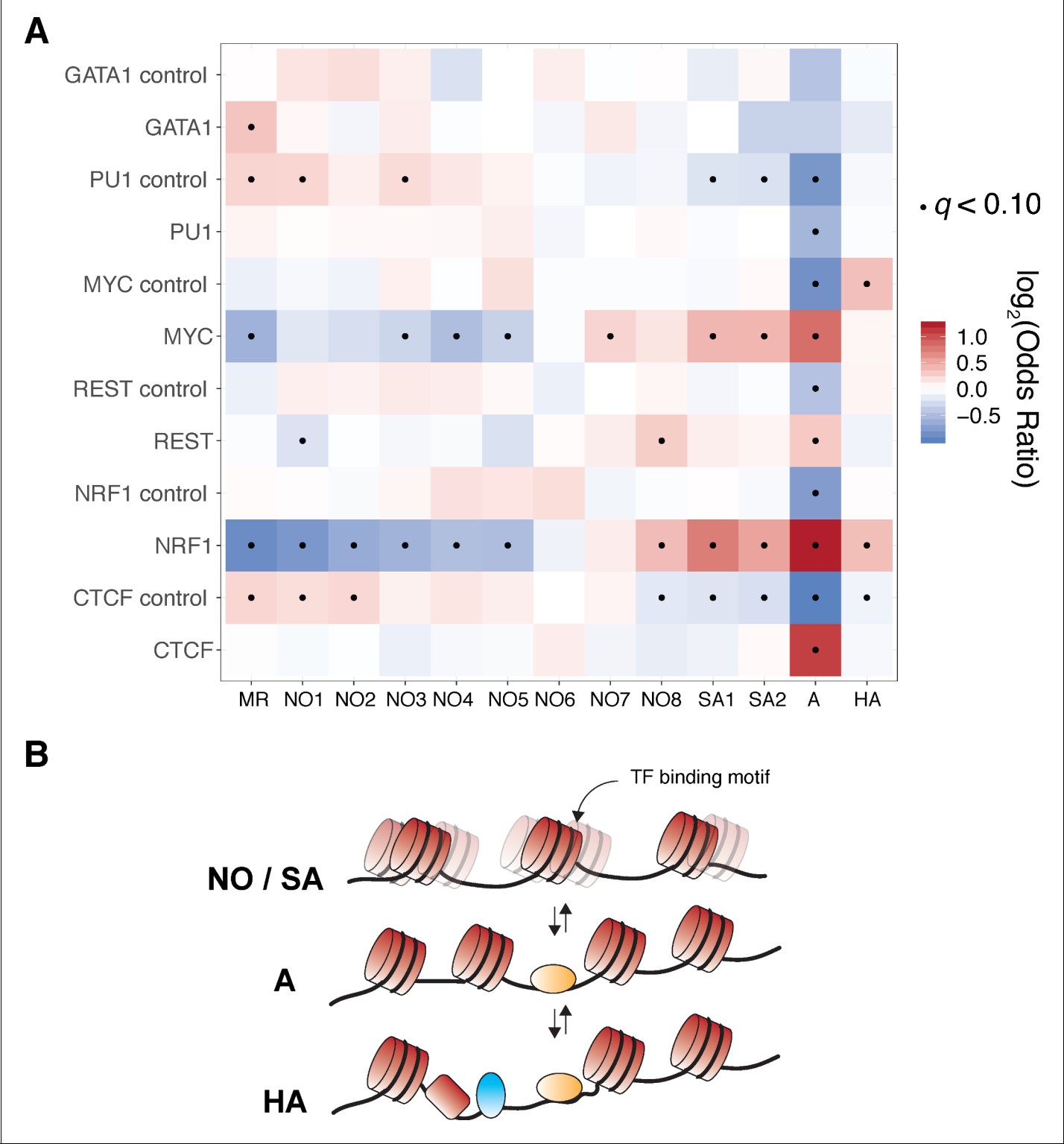

**Figure 5.** TF-centric clusters exhibit significantly different usage of specific 'registers' of nucleosome positioning with respect to predicted TF-binding sites. (**A**) We performed Fisher's exact tests to determine relative enrichment and depletion of each cluster for each transcription factor surveyed in *Figure 4*. Cluster 'A' is consistently depleted across control molecules but enriched across molecules containing bona fide transcription factor binding motifs, suggesting that the clusters identified in this study are biologically relevant. Fishers Exact test odds ratios are plotted in heatmap form and all enrichment tests that are statistically significant under a false discovery rate of 10% (q < 0.1) are marked with a black dot. (**B**) Our data may be explained by the Widom 'site exposure' model in vivo. Transcription factor binding motifs are stochastically exposed as nucleosomes toggle between

*Figure 5 continued on next page*

*Figure 5 continued*

multiple 'registers' as seen in *Figure 4M* (states NO and SA). Transcription factor binding perhaps enforces a favorable nucleosome register (state A), which can then seed hyper-accessible states/further TF-DNA interactions (state HA).

The online version of this article includes the following figure supplement(s) for figure 5:

**Figure supplement 1.** Reproducibility of transcription factor enrichment analyses.

molecules, a finding consistent with the expected irregularity of active promoter oligonucleosomes. We note that all these measured parameters would be unattainable using any existing biochemical method and that these preliminary findings argue against the abundance of homogeneous oligonucleosome structures in either heterochromatic or euchromatic nuclear regions.

On first glance, our data appear to run counter to previous observations demonstrating that epigenomic domains can be delineated by differences in bulk nucleosome positioning as measured by nuclease digestion. One possible explanation for this is that epigenomic domains subtly, but significantly, vary in their relative composition of distinct oligonucleosome patterns, and the resulting average of these differences is the signal captured in MNase-Southern and other cleavage-based measurements. We tested this hypothesis by constructing a series of statistical tests to determine whether each of the seven defined oligonucleosome patterns were significantly enriched or depleted across chromatin domains and matched control regions (*Figure 6C*; reproducibility analyses summarized in *Figure 6—figure supplement 1*). Our results suggest that chromatin domains are demarcated by their relative usage of specific oligonucleosome patterns. Consistent with expectations, active chromatin marked by H3K4me3 and H3K4me1 are punctuated by a mixture of irregular oligonucleosome patterns (namely, clusters IRL and IR170). For transcription elongation associated H3K36me3 decorated chromatin, both short-read mapping in human and long-read bulk array regularity mapping in *D. melanogaster* have suggested relatively short, regular nucleosome repeat lengths (*Baldi et al., 2018*; *Valouev et al., 2011*). Our data partially corroborate this finding in human K562 cells: H3K36me3-domains are punctuated by irregular IRS oligoncleosome patterns (Fisher's Exact Odds Ratio [O.R.]=1.13; $q = 1.71E-50$) and regular, short NRL172 patterns (O.R. = 1.39; $q = 3.69E-170$).

Our assay also allows us to assess compositional biases in heterochromatic domains. Short-read-based human studies and classical MNase mapping of constituve heterochromatin have suggested that H3K9me3-decorated chromatin harbor (i) long nucleosome repeat lengths on average, and (ii) are highly regular. These estimates are susceptible to artifacts, as heterochromatic nucleosomes are expected to be both strongly phased and weakly positioned. Our data partially disagree with prior estimates—across both H3K9me3 and Satellite molecules we observe enrichment for irregular IRS nucleosome conformers (Satellite O.R. = 1.13; $q = 5.71E-11$; H3K9me3 O.R. = 1.35; $q = 3.95E-23$). Still, these enriched conformers were accompanied by enrichment for regular NRL172 oligonucleosome patterns for both states (Satellite O.R. = 1.61; $q = 5.25E-80$; H3K9me3 O.R. = 1.23; $q = 3.86E-6$). These analyses demonstrate that prior NRL estimates by short-read sequencing may have been confounded by in vivo heterogeneity in nucleosome positions, that heterochromatic nucleosome conformations can be both irregular and diverse, and finally, highlight the value of SAMOSA for accurately studying nucleosome structure in heterochromatin.

Taken as a whole, our data suggest two fundamental properties of human epigenomic domains: first, epigenomic domains are comprised of a diverse array of oligonucleosome patterns varying substantially in intrinsic regularity and average distance between regularly spaced nucleosomes; second: epigenomic domains are demarcated by their usage of these oligonucleosome patterns. We find that all epigenomic states are characterized by a diverse mixture of oligonucleosomal conformers— many conformational states are neither significantly depleted nor enriched with respect to all molecules surveyed, further hinting at the diverse composition of chromatin domains genome-wide.

## Discussion

Here, we present the SAMOSA, a method for resolving nucleosome-DNA interactions using the EcoGII adenine methyltransferase and PacBio single-molecule real-time sequencing. Our approach has multiple advantages over existing methyltransferase-based sequencing approaches: first, by using a

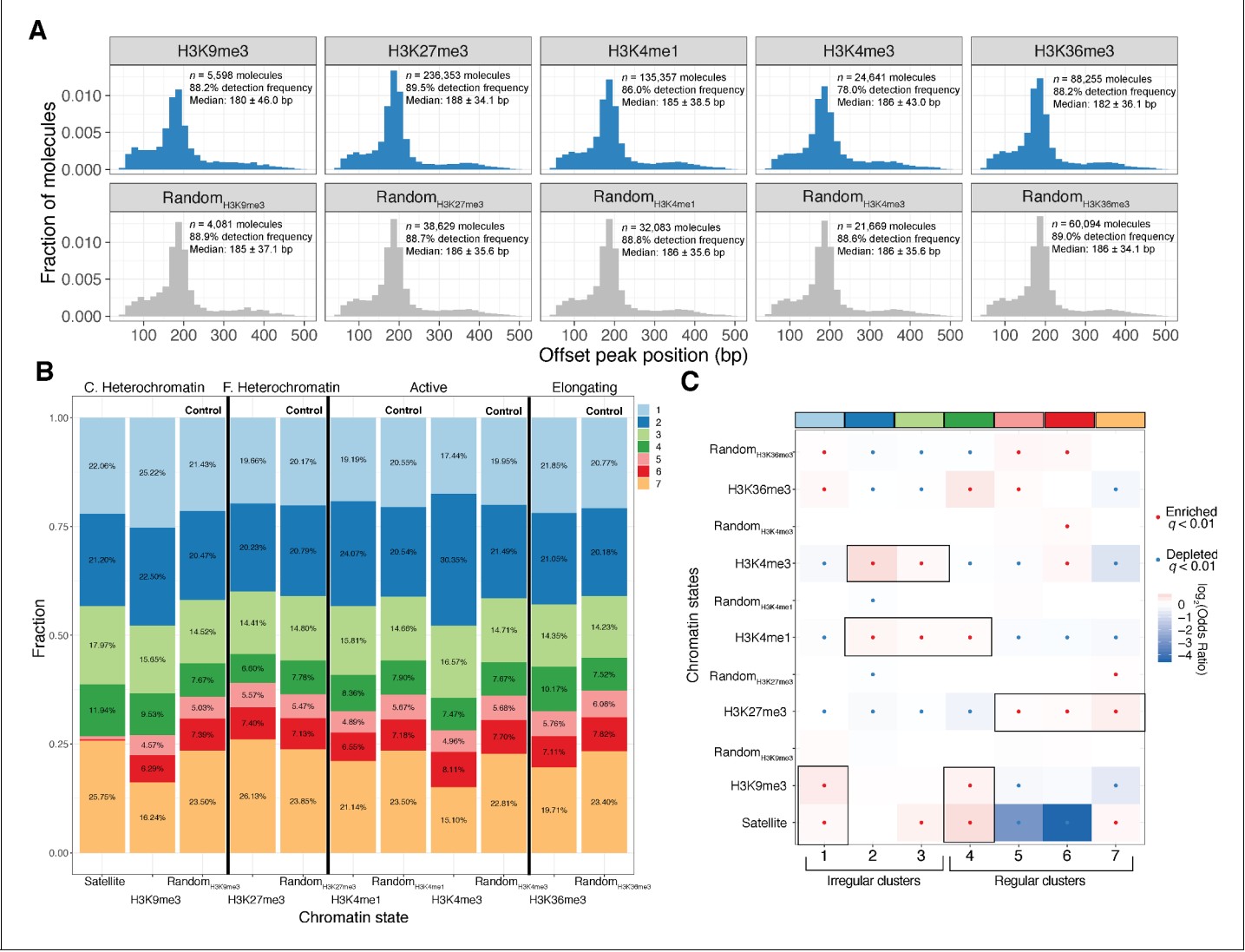

**Figure 6.** Human epigenomic states are punctuated by specific oligionucleosome patterns. A) Histogram representations of the estimated single-molecule NRLs for five different epigenomic domains compared to control sets of molecules matched for GC and repeat content. Inset: Numbers of molecules plotted, median NRL estimates with associated median absolute deviations, and the percent of molecules where a peak could not be detected. (B) Stacked bar chart representation of the relative composition of each epigenomic domain with respect to the seven clusters defined in *Figure 3*. C. Heterochromatin: constitutive heterochromatin; F. Heterochromatin: facultative heterochromatin. (C). Heatmap of enrichment test results to determine nucleosome conformers that are enriched or depleted for each chromatin state. Tests qualitatively appearing to be chromatin-state specific are highlighted with a black box. Significant tests following multiple hypothesis correction marked with a black dot. Fisher's Exact Test was used for all comparisons.

The online version of this article includes the following figure supplement(s) for figure 6:

**Figure supplement 1.** Reproducibility analysis of chromatin state analyses.

**Figure supplement 2.** Reanalysis of the Fiber-seq data of Stergachis et al validates SAMOSA-based findings of our initial submission.

**Figure supplement 3.** Satellite-specific chromatin analyses reveal differences between fibre-usage across H3K9me3-positive and H3K9me3-negative satellite repeats.

relatively nonspecific methyltransferase, we avoid the primary sequence biases associated with GpC/CpG methyltransferase footprinting methods; second, by natively detecting modifications using the single-molecule real-time sequencer, we reduce enzymatic sequence bias and avoid sample damage associated with sodium bisulphite conversion; finally, and most importantly, our approach unlocks the study of protein-DNA interactions at length-scales previously unallowed by Illumina sequencing.

Our study does have limitations. While the current SAMOSA protocol enriches fragments ranging from ~500 bp to ~ 2 kb in size, high-quality PacBio CCS sequencing is compatible with fragments ranging from 10 to 15 kbp. We anticipate that with further optimization (e.g. optimization of digestion conditions), SAMOSA will be applicable to longer arrays, enabling kilobase-domain-scale study of single-molecule oligonucleosome patterning. Indeed, our preliminary SAMOSA experiments varying digestion conditions demonstrate the feasibility of such variations (*Figure 2—figure supplement 1*). Second, our approach involves methylating fibres following solubilization of oligonucleosomal fragments, and is thus unlikely to capture protein-DNA interactions weaker or more transient than the stable nucleosome-DNA interaction. Such transient interactions could be captured in future work by modifying the protocol to footprint nuclei prior to MNase-solubilization. Third, our proof-of-concept was performed in unsynchronized K562 cells, and thus we cannot yet address the contribution of a biological process like the cell cycle to the observed heterogeneity. Finally, as a proof-of-concept our approach falls short of generating a high-coverage reference map of the K562 epigenome; as sequencing costs for PacBio decrease and sequence-enrichment technologies (*e.g.* CRISPR-based enrichment *Ebbert et al., 2018*; SMRT-ChIP [*Wu et al., 2016*]) for the platform mature, SAMOSA may routinely be used to generate reference datasets with hundred-to-thousand-fold single-molecular coverage of genomic sites of interest.

Our data confirms that the human epigenome is made up of a diverse array of oligonucleosome patterns, including highly regular arrays of varying nucleosome repeat lengths, and irregular arrays where nucleosomes are positioned without a detectable periodic signature (*Baldi et al., 2020*). Our results broadly agree with a recent approach employing electron tomography to map the in situ structure of mammalian nuclei, which found chromatin to be highly heterogeneous at the length scale of multi-nucleosome interactions, and failed to detect evidence of a 30 nm fibre or other homogeneous higher order compaction states (*Ou et al., 2017*). At the sequencing depth presented here, these oligonucleosome patterns significantly, if subtly, vary across different epigenomic domains. Surprisingly, we find that both mappable (H3K9me3 ChIP-seq peaks) and unmappable (human satellite sequence) constitutive heterochromatin are enriched for irregular oligonucleosome patterns in addition to expected regular arrays—the presence of these irregular fibres may have been previously missed due to an understandable reliance on bulk averaged methods (e.g. MNase-Southern) for studying constitutive heterochromatin. This is strongly supported by orthogonal analysis of heterochromatin-spanning K562 reads generated using the recently published, conceptually similar Fiber-seq method (*Stergachis et al., 2020*), which also reveal that H3K9me3 domains are enriched for irregular chromatin fibres (*Figure 6—figure supplement 2*). Given the robustness of this finding, it is tempting to speculate that this irregularity may be linked to the dynamic restructuring of heterochromatic nucleosomes by factors like HP1 (*Sanulli et al., 2019*), which may promote phase-separation of heterochromatin. While stratification of analyzed satellite sequences into H3K9me3-decorated alpha/beta, and H3K9me3-free gamma satellite (*Kim et al., 2009*) provides correlative support for this notion (*Figure 6—figure supplement 3*), future studies combining SAMOSA with cellular perturbation of heterochromatin-associated factors are necessary to directly address this possibility.

More generally, future work employing our technique must focus on questioning the biological significance of this global heterogeneity: for example, is the fraction of stochastically accessible transcription factor binding sites (i.e. motif 'site exposure' frequency [*Ahmad and Henikoff, 2001*; *Polach and Widom, 1995*]) important for TF-DNA binding in nucleosome-occluded genomic regions? What is the interplay between transcription factor 'pioneering' and stochastic site accessibility? What are the global roles of ATP-dependent chromatin remodeling enzymes (i.e. SWI/SNF; ISWI; INO80; CHD) in maintaining these patterns genome-wide (*Brahma and Henikoff, 2020*)? Our approach also unlocks a set of conceptual questions regarding the nature of chromatin secondary structure. Significant genome-wide efforts have revealed that metazoan epigenomes are punctuated by regions of concerted histone modification and subnuclear positioning (*ENCODE Project Consortium, 2012*; *Filion et al., 2010*), but approaches for studying the distribution of oligonucleosomal patterns associated within these same regions are lacking. Given recent work suggesting that NRLs can specify the ability of nucleosomal arrays to phase separate (*Gibson et al., 2019*), it is likely that SAMOSA and similar assays may provide an important bridge between in vitro biochemical observations of chromatin and in vivo genome-wide 'catalogs' of oligonucleosome patterning.

SAMOSA adds to the growing list of technologies that use high-throughput single-molecule sequencing to explore the epigenome (*Baldi et al., 2018*; *Lee et al., 2019*; *Shipony et al., 2020*; *Wang et al., 2019*; *Stergachis et al., 2020*). We foresee the broad applicability of this and similar approaches to dissect gene regulatory processes at previously intractable length-scales. Our approach and associated analytical pipelines demonstrate the versatility of high-throughput single-molecule sequencing—namely the ability to cluster single-molecules in an unsupervised manner to uncover molecular states previously missed by short-read approaches. Our analytical approach bears many similarities to methods used in single-cell analysis, and indeed many of the technologies and concepts typically used for single-cell genomics (*Trapnell, 2015*) (e.g. clustering; trajectory analysis) will likely have value when applied to single-molecule epigenomic assays. Our approach also follows in the footsteps of multi-omic Illumina assays like NoME-seq and MapIT, representing the first of what we anticipate will be many 'multi-omic' third-generation sequencing assays. As third-generation sequencing technologies advance, it will likely become possible to encode multiple biochemical signals on the same single-molecules, thus enabling causal inference of the logic and ordering of biochemical modifications on single chromatin templates.

## Materials and methods

### Preparation of nonanucleosome arrays via salt-gradient dialysis

The nonanucleosome DNA in a plasmid was purified by Gigaprep (Qiagen) and the insert was digested out with EcoRV, ApaLI, XhoI and StuI. The insert was subsequently purified using a Sephacryl S1000 super fine gel flitration (GE Healthcare). Histones were purified and octamer was assembled as previously described (*Luger et al., 1999*). To assemble the arrays, the nonanucleosome DNA was mixed with octamer and supplementing dimer, then dialyzed from high salt to low salt (*Lee and Narlikar, 2001*). EcoRI sites engineered in the linker DNA between the nucleosomes, and digestion by EcoRI was used to assess the quality of nucleosome assembly.

### SAMOSA on nonanucleosomal chromatin arrays

For the chromatin arrays, 1.5 µg of assembled array was utilized as input for methylation reactions with the non-specific adenine EcoGII methyltransferase (New England Biolabs, high concentration stock; 2.5E4U/mL). For the naked DNA controls, 2 µg of DNA was utilized as input for methylation reactions. Methylation reactions were performed in a 100 µL reaction with Methylation Reaction buffer (1X CutSmart Buffer,1 mM S-adenosyl-methionine (SAM, New England Biolabs)) and incubated with 2.5 µL EcoGII at 37°C for 30 min. SAM was replenished to 6.25 mM after 15 min. Unmethylated controls were similarly supplemented with Methylation Reaction buffer, minus EcoGII and replenishing SAM, and the following purification conditions. To purify DNA, the samples were all subsequently incubated with 10 uL Proteinase K (20 mg/mL) and 10 µL 10% SDS at 65°C for a minimum of 2 hr up to overnight. To extract the DNA, equal parts volume of Phenol-Chloroform was added and mixed vigorously by shaking, spun (max speed, 2 min). The aqueous portion was carefully removed and 0.1x volumes of 3M NaOAc, 3 µL of GlycoBlue and 3x volumes of 100% EtOH were added, mixed gently by inversion, and incubated overnight at −20°C. Samples were then spun (max speed, 4°C, 30 min), washed with 500 µL 70% EtOH, air dried and resuspended in 50 µuL EB. Sample concntration was measured by Qubit High Sensitivity DNA Assay.

### Preparation of in vitro SAMOSA SMRT libraries

The purified DNA from nonanucleosome array and DNA samples were used in entirety as input for PacBio SMRTbell library preparation (~1.5–2 µg). Preparation of libraries included DNA damage repair, end repair, SMRTbell ligation, and Exonuclease according to manufacturer's instruction. After Exonuclease Cleanup and a double 0.8x Ampure PB Cleanup, sample concentration was measured by Qubit High Sensitivity DNA Assay (1 µL each). To assess for library quality, samples (1 µL each) were run on an Agilent Bioanalyzer DNA chip. Libraries were sequenced on either Sequel I or Sequel II flow cells (UC Berkeley QB3 Genomics). Sequel II runs were performed using v2.0 sequencing chemistry and 30 hr movies.

## Cell lines and cell culture

K562 cells (ATCC) were grown in standard media containing RPMI 1640 (Gibco) supplemented with 10% Fetal Bovine Serum (Gemini, Lot#A98G00K) and 1% Penicillin-Streptomycin (Gibco). Cell lines were regularly tested for mycoplasma contamination and confirmed negative with PCR (NEB Neb-Next Q5 High Fidelity 2X Master Mix).

## Isolation of nuclei, MNase digest, and overnight dialysis

100E6 K562 cells were collected by centrifugation (300x**g**, 5 min), washed in ice cold 1X PBS, and resuspended in 1 mL Nuclear Isolation Buffer (20 mM HEPES, 10 mM KCl, 1 mM $MgCl_2$, 0.1% Triton X-100, 20% Glycerol, and 1X Protease Inhibitor (Roche)) per 5–10 e6 cells by gently pipetting 5x with a wide-bore tip to release nuclei. The suspension was incubated on ice for 5 min, and nuclei were pelleted (600xg, 4°C, 5 min), washed with Buffer M (15 mM Tris-HCl pH 8.0, 15 mM NaCl, 60 mM KCl, 0.5 mM Spermidine), and spun once again. Nuclei were resuspended in 37°C pre-warmed Buffer M supplemented with 1 mM $CaCl_2$ and distributed into two 1 mL aliquots. For digestion, micrococcal nuclease from *Staphylococcus aureus* (Sigma, reconstituted in $ddH_2O$, stock at 0.2 U/uL) was added at 1U per 50E6 nuclei, and nuclei were digested for 1 min. at 37°C. EGTA was added to 2 mM immediately after 1 min to stop the digestion and incubated on ice. For nuclear lysis and liberation of chromatin fibres, MNase-digested nuclei were collected (600xg, 4°C, 5 min) and resuspended in 1 mL per 50E6 nuclei of Tep20 Buffer (10 mM Tris-HCl pH 7.5, 0.1 mM EGTA, 20 mM NaCl, and 1X Protease Inhibitor (Roche) added immediately before use) supplemented with 300 µg/mL of Lysolethicin (L-$\alpha$-Lysophosphatidylcholine from bovine brain, Sigma, stock at 5 mg/mL) and incubated at 4°C overnight. To remove nuclear debris the next day, dialyzed samples were spun (12,000xg, 4°C, 5 min) and the soluble chromatin fibres present in the supernatant were collected. Sample concentration was measured by Nanodrop. SAMOSA experiments with variable digestion conditions were performed as above, except temperature (37°C vs. 4°C) and time (1 min vs. 10 min vs. 60 min) were varied, starting cell counts were increased to 200E6 for prepared nuclei for varied condition experiments, and gTube spins were omitted.

## SAMOSA on K562-derived oligonucleosomes

Dialyzed chromatin was utilized as input (1.5 µg) for methylation reactions with the non-specific adenine EcoGII methyltransferase (New England Biolabs, high concentration stock 2.5e4U/mL). Reactions were performed in a 200 µL reaction with 1X CutSmart Buffer and 1 mM S-adenosyl-methionine (SAM, New England Biolabs) and incubated with 2.5 µL enzyme at 37°C for 30 min. SAM was replenished to 6.25 mM after 15 min. Non-methylation controls were similarly supplemented with Methylation Reaction buffer, minus EcoGII and replenishing SAM, and purified by the following conditions. To purify all DNA samples, reactions were incubated with 10 µL of RNaseA at room temperature for 10 min, followed by 20 uL Proteinase K (20 mg/mL) and 20 uL 10% SDS at 65°C for a minimum of 2 hr up to overnight. To extract the DNA, equal parts volume of Phenol-Chloroform was added and mixed vigorously by shaking, spun (max speed, 2 min). The aqueous portion was carefully removed and 0.1x volumes of 3M NaOAc, 3 µL of GlycoBlue and 3x volumes of 100% EtOH were added, mixed gently by inversion, and incubated overnight at −20°C. Samples were then spun (max speed, 4°C, 30 min), washed with 500 µL 70% EtOH, air dried and resuspended in 50 µL EB. Sample concentration was measured by Qubit High Sensitivity DNA Assay. Naked DNA Positive methylation controls were collected from aforementioned non-methylated controls post-purification (25 µL, ~500 ng), methylated with EcoGII as previously stated, and purified again by the following conditions.

## Preparation of in vivo SAMOSA SMRT libraries

Purified DNA from MNase-digested K562 chromatin oligonucleosomes (methylated, non-methylated control, purified then methylated) were briefly spun in a Covaris G-Tube (3380xg, 1 min) in efforts to shear gDNA uniformly to 10 kB prior PacBio library preparation. The input concentration was approximately 575 ng for methylated and non-methylated samples, and approximately 320 ng for purified then methylated samples. Samples were concentrated with 0.45x of AMPure PB beads according to manufacturer's instructions. The entire sample volume was utilized as input for subsequent steps in library preparation, which included DNA damage repair, end repair, SMRTbell ligation, and Exonuclease cleanup according to manufacturer's instructions. For SMRTbell ligations,

unique PacBio SMRT-bell adaptors (100 µM stock) were annealed to a 20 µM working stock in 10 mM Tris-HCl pH 7.5 and 100 mM NaCl in a thermocycler (85℃ 5 min, RT 30 s, 4℃ hold) and stored at −20℃ for long-term storage. After exonuclease cleanup and double Ampure PB cleanups (0.45X), the sample concentrations were measured by Qubit High Sensitivity DNA Assay (1 µL each). To assess for size distribution and library quality, samples (1 uL each) were run on an Agilent Bioanalyzer DNA chip. Libraries were sequenced on Sequel II flow cells (UC Berkeley QB3 Genomics Core). In vivo data were collected over three 30 hr Sequel II movie runs; the first with a 2 hr pre-extension time and the second two with a 0.7 hr pre-extension time.

## Data analysis

All raw data will be made available at GEO Accession GSE162410; processed data is available at Zenodo (https://doi.org/10.5281/zenodo.3834705). All scripts and notebooks for reproducing analyses in the paper are available at https://github.com/RamaniLab/SAMOSA (*Abdulhay, 2020*; copy archived at swh:1:rev:208027064183d042adede691b935cad9e79106a3).

We apply our method to two use cases in the paper, and they differ in the computational workflow to analyze them. The first is for sequencing samples where every DNA molecule should have the same sequence, which is the case for our in vitro validation experiments presented in *Figure 1*. The second use case is for samples from cells containing varied sequences of DNA molecules. We will refer to the first as homogeneous samples, and the second as genomic samples. The workflow for genomic samples will be presented first in each sections, and the deviations for homogeneous samples detailed at the end.

500U hia5 K562 Fiber-seq data from *Stergachis et al., 2020* were downloaded using Google Cloud Services via SRA accession SRP252718 and processed as below.

## Sequencing read processing

Sequencing reads were processed using software from Pacific Biosciences. The following describes the workflow for genomic samples:

### Demultiplex reads

Reads were demultiplexed using lima. The flag '–same' was passed as libraries were generated with the same barcode on both ends. This produces a BAM file for the subreads of each sample.

### Generate circular consensus sequences (CCS)

CCS were generated for each sample using ccs (*Travers et al., 2010*). Default parameters were used other than setting the number of threads with '-j'. This produces a BAM file of CCS.

### Align CCS to the reference genome

Alignment was done using pbmm2 (*Li, 2016*), and run on each CCS file, resulting in BAM files containing the CCS and alignment information.

### Generate missing indices

Our analysis code requires pacbio index files (.pbi) for each BAM file. 'pbmm2' does not generate index files, so missing indices were generated using 'pbindex'.

For homogeneous samples, replace step three with this alternate step 3.

### Align subreads to the reference genome

pbmm2 was run on each subreads BAM file (the output of step 1) to align subreads to the reference sequence, producing a BAM file of aligned subreads.

## Sample reference preparation

Our script for analyzing samples relies on a CSV file input that contains information about each sample, including the locations of the relevant BAM files and a path to the reference genome. The CSV needs a header with the following columns: **index**: Integer indices for each sample. We write the table using 'pandas' '.to_csv' function, with parameters 'index = True, index_label='index'' **cell**: A unique name for the SMRT cell on which the sample was sequenced **sampleName**: The name of the

sample **unalignedSubreadsFile**: This will be the file produced by step one above. This should be an absolute path to the file.

### ccsFile
This is the file produced by step two above **alignedSubreadsFile**: This is the file produced by the alternate step three above. It is required for homogeneous samples but can be left blank for genomic samples.

### alignedCcsFile
This is the file produced by step three above. It is required for genomic samples but can be left blank for homogeneous samples.

### Reference
The file of the reference genome or reference sequence for the sample.

## Extracting IPD measurements and calling methylation
The script extractIPD.py accesses the BAM files, reads the IPD values at each base and uses a gaussian mixture model to generate posterior probabilities of each adenine being methylated. extractIPD takes two positional arguments. The first is a path to the above sample reference CSV file. The second is a specification for which sample to run on. This can be either an integer index value, in which case extractIPD will run on the corresponding row. Alternatively it can be a string containing the cell and sampleName, separated by a period. Either way extractIPD will run on the specified sample using the paths to the BAM files contained within the CSV.

extractIPD produces the following three output files when run on genomic samples: **processed/ onlyT/{cell}_{sampleName}_onlyT_zmwinfo.pickle:** This file is a 'pandas' dataframe stored as a pickle, and can be read with the 'pandas.read_pickle' function. This dataframe contains various information about each individual ZMW.

### Processed/onlyT/{cell}_{sampleName}_onlyT.pickle
This file contains the normalized IPD value at every thymine. The data is stored as a dictionary object. The keys are the ZMW hole numbers (stored in the column 'zmw' in the zmwinfo dataframe), and the values are numpy arrays. The arrays are 1D with length equal to the length of the CCS for that molecule. At bases that are A/T, there will be a normalized IPD value. Each G/C base and a few A/T bases for which an IPD value couldn't be measured will contain NaN.

### Processed/binarized/{cell}_{sampleName}_bingmm.pickle
This file contains the posterior probability of each adenine being methylated. The data format is identical to the _onlyT.pickle file above, except the numpy array contains values between 0 and 1, where the higher values indicate a higher confidence that the adenine is methylated.

When run on homogeneous samples the following output files are alternately produced: **processed/onlyT/{cell}_{sampleName}_onlyT.npy:** This numpy array has a column for every base in the reference sequence, and a row for each DNA molecule that passes the filtering threshold. A normalized IPD value is stored for each adenine that could be measured at A/T bases, other bases are NaN.

### Processed/binarized/{cell}_{sampleName}_bingmm.npy
This numpy array is the same shape as the _onlyT.npy file above. The values are posterior probabilities for an adenine being methylated, ranging from 0 to 1.

## Dyad calling on in vitro methylated chromatin arrays
Nucleosome positions were predicted in nonanucleosomal array data by taking a 133 bp wide rolling mean across the molecule, and finding each local minimum peak at least 147 bp apart from each other.

### k-*mer* analyses of negative and positive control experiments

To investigate the role of sequence context in our methylation calls, we examined the distribution of normalized IPD values for our in vitro negative and positive controls. We binned the adenines by sequence context using two base pairs on the 5' side of the template base and five base pairs on the 3' side. These bases were previously found to have the strongest influence on IPD value [ref in revision response google doc]. We combined both replicates for negative and positive controls and plotted a heatmap where each row is a sequence context and the color intensity is the histogram counts of molecules with a normalized IPD value in that bin. Negative control, positive control, and both combined were each plotted. K-mer contexts were sorted by their mean normalized IPD in the combined set. The sequence contexts were separately plotted.

### In vivo analyses

We smooth the posterior probabilities calculated in the paper to account for regions with low local A/T content and generally denoise the single-molecule signal. For in vitro analyses, we smooth the calculated posterior probabilities using a 5 bp rolling mean. For all in vivo analyses in the paper that involve calculation of single-molecule autocorrelograms, averaging over multiple templates, and visualizing individual molecules, we smooth posteriors with a 33 bp rolling mean. For all autocorrelation calculations we ignore regions where compared lengths would be unequal; this has the effect of rendering the returned autocorrelogram exactly 0.5 * the input length.

## Averages of the modification signal across the first 1 kb of K562 oligonucleosomes

We took all molecules at least 500 nt in length and concatenated all of the resulting matrices from each of the four separate samples/runs, and then plotted the NaN-sensitive mean over the matrix as a function of distance along the molecule.

## Clustering analysis of all chromatin molecules >= 500 bp in length

We used Leiden clustering cluster all molecules in our dataset passing our lower length cutoff. Resolution and n_neighbors were manually adjusted to avoid generating large numbers of very small clusters (i.e. <100 molecules). All parameters used for plotting figures in the paper are recapitulated in the Jupyter notebook. Our clustering strategy was as follows: first, we smoothed raw signal matrices with a 33 bp NaN-sensitive running mean. We next computed the autocorrelation function for each molecule in the matrix, using the full length of the molecule up to 1000 bp. We then used Scanpy (*Wolf et al., 2018*) to perform Leiden clustering on the resulting matrix. We visualized the resulting cluster averages with respect to the average autocorrelation function, and with respect to averaged modification probabilities for each cluster. For a subset of clusters we also randomly sampled 500–5000 molecules to directly visualize in the paper.

## Computing single-molecule autocorrelograms and estimating NRLs on single molecules

We computed single-molecule autocorrelograms and discovered peaks on these autocorrelograms as follows: for each molecule, we used the scipy (*Virtanen et al., 2020*) find_peaks function to in the computed autocorrelogram and annotated the location of that peak. We also kept track of the molecules where find_peaks could not detect a peak using the given parameters, which we optimized manually by modifying peak height/width to detect peaks on the averaged autocorrelograms. In our hands, these parameters robustly detect peaks between 180 and 190 bp in auto-correlogram averages, consistent with the expected bulk NRL in K562 cells (analyses by A Rendeiro; zenodo.org/record/3820875). For each collection of single-molecule autocorrelogram peaks we computed the median, the median absolute deviation, and visualized the distribution of peak locations as a histogram.

### TF-binding motif analyses and enrichment tests

K562 TF-binding sites were predicted as in *Ramani et al., 2019*. Briefly, we downloaded IDR-filtered ENCODE ChIP-seq peaks for CTCF, NRF1, REST, c-MYC, PU.1, and GATA1, and then used FIMO (*Bailey et al., 2009*) to predict TF binding sites within these peaks using CISTROME PWM

definitions for each transcription factor. For MNase-cleavage analyses, we plotted the abundance of MNase cuts (two per molecule) with respect to TF binding sites and plotted these as number of cleavages per molecules sequenced. To examine modification probabilities around TF-binding sites, we wrote a custom script (zmw_selector.py) to find the ZMWs that overlap with features of interest (e.g. transcription factor binding sites). We extracted all ZMWs where a portion of the read alignment falls within 1 kb of a given feature, and annotated the position of the alignment starts, ends, and strand with respect to the feature. We then used these coordinates and strand information to extract all modification signal falling within a 500 bp window centered at each TF binding site. For control sites, we used the gkmSVM package (*Ghandi et al., 2016*) to find GC-/repeat content matched genomic regions for each peakset. We constructed a series of enrichment tests (Fisher's Exact) to determine odds ratios/p values to find specific cluster label–transcription factor pairs that were enriched with respect to the total set of all labeled molecules. Finally, we used the Storey q-value package (*Storey and Tibshirani, 2003*) to correct for the number of Fisher's exact tests performed.

### Enrichment tests for chromatin states

We used a custom python script (zmw_selector_bed.py) or directly scanned for satellite-containing CCS reads (see below) to extract molecules that fall within ENCODE-defined chromatin states/pertain to human major satellite sequences. We then used a Python dictionary linking ZMW IDs to indices along the total matrix of molecules to link Cluster IDs and chromatin states. Finally, we constructed a series of enrichment tests (Fisher's Exact) to determine odds ratios/p values to find specific cluster label-chromatin state pairs that were enriched with respect to the total set of all labeled molecules. We then used the Storey q-value package to correct for the number of Fisher's exact tests performed. Control molecules were drawn as above, using the gkmSVM package to find GC/repeat content matched genomic regions for each peakset.

### Selection of satellite-containing reads

Circular consensus reads with minimum length of 1 kb bearing satellites were identified using BLAST searching against a database containing DFAM (*Hubley et al., 2016*) consensus sequences for alpha (DF0000014.4, DF0000015.4, DF0000029.4), beta (DF0000075.4, DF0000076.4, DF0000077.4, DF0000078.4, DF0000079.4), and gamma (DF0000148.4, DF0000150.4, DF0000152.4) satellites using blastn with default parameters. Satellite containing reads were further filtered such that they contained at minimum two hits to satellite consensus sequences and matches spanned at least 50% of the consensus sequence. These labels were then used to separate out sequences for the analyses presented in *Figure 6—figure supplement 3*.

## Acknowledgements

The authors thank Daniele Canzio (UCSF), Hiten Madhani (UCSF), Srinivas Ramachandran (CU Denver), and Christopher Weber (Stanford) for helpful discussions and comments on the manuscript. The authors thank Shana McDevitt, Robert Munch, and the UC Berkeley Vincent J Coates Genomics Sequencing Laboratory for assisting with PacBio sequencing.

## Additional information

### Competing interests

Geeta J Narlikar: Reviewing editor, *eLife*. Jason G Underwood: JGU is an employee of Pacific Biosciences, Inc and holds stock in this company. The other authors declare that no competing interests exist.

### Funding

| Funder | Grant reference number | Author |
|---|---|---|
| Sandler Foundation | | Vijay Ramani |
| American Cancer Society | | Laura J Hsieh |

| National Institutes of Health | R01GM123977 | Hani Goodarzi |
|---|---|---|
| National Institutes of Health | R35GM127020 | Geeta J Narlikar |

The funders had no role in study design, data collection and interpretation, or the decision to submit the work for publication.

### Author contributions
Nour J Abdulhay, Conceptualization, Formal analysis, Investigation, Methodology, Writing - original draft, Writing - review and editing; Colin P McNally, Conceptualization, Data curation, Software, Formal analysis, Investigation, Methodology, Writing - original draft, Writing - review and editing; Laura J Hsieh, Investigation, Methodology; Sivakanthan Kasinathan, Software, Investigation, Methodology; Aidan Keith, Investigation, Methodology, Writing - review and editing; Laurel S Estes, Mehran Karimzadeh, Software, Formal analysis, Writing - review and editing; Jason G Underwood, Conceptualization; Hani Goodarzi, Geeta J Narlikar, Supervision, Funding acquisition, Investigation, Writing - review and editing; Vijay Ramani, Conceptualization, Data curation, Software, Formal analysis, Supervision, Funding acquisition, Investigation, Methodology, Writing - original draft, Writing - review and editing

### Author ORCIDs
Mehran Karimzadeh ![ORCID] http://orcid.org/0000-0002-7324-6074
Geeta J Narlikar ![ORCID] http://orcid.org/0000-0002-1920-0147
Vijay Ramani ![ORCID] https://orcid.org/0000-0003-3345-5960

### Decision letter and Author response
Decision letter https://doi.org/10.7554/eLife.59404.sa1
Author response https://doi.org/10.7554/eLife.59404.sa2

# Additional files

### Supplementary files
• Transparent reporting form

### Data availability
All raw data are available at GEO Accession GSE162410; processed data is available at Zenodo (https://doi.org/10.5281/zenodo.3834705). All scripts and notebooks for reproducing analyses in the paper are available at https://github.com/RamaniLab/SAMOSA (copy archived at https://archive.softwareheritage.org/swh:1:rev:208027064183d042adede691b935cad9e79106a3/).

The following datasets were generated:

| Author(s) | Year | Dataset title | Dataset URL | Database and Identifier |
|---|---|---|---|---|
| Abdulhay NJ, McNally CP, Hsieh LJ, Kasinathan S, Keith A, Estest LS, Karimzadeh M, Underwood JG, Goodarzi H, Narlikar GJ, Ramani V | 2020 | Massively multiplex single-molecule oligonucleosome footprinting | https://zenodo.org/record/3834706 | Zenodo , 10.5281/zenodo.3834706 |
| Colin M, Nour A, Vijay R | 2020 | Massively multiplex single-molecule oligonucleosome footprinting | https://www.ncbi.nlm.nih.gov/geo/query/acc.cgi?acc=GSE162410 | NCBI Gene Expression Omnibus, GSE162410 |

The following previously published datasets were used:

| Author(s) | Year | Dataset title | Dataset URL | Database and Identifier |
|---|---|---|---|---|
| ENCODE Consortium | 2011 | K562 H3K27me3 ChIP | https://www.encodeproject.org/experiments/ENCSR000EWB/ | ENCODE, ENCFF031FSF |
| ENCODE Consortium | 2011 | K562 H3K36me3 ChIP | https://www.encodeproject.org/experiments/ENCSR000DWB/ | ENCODE, ENCFF631VWP |
| ENCODE Consortium | 2016 | K562 H3K4me3 ChIP | https://www.encodeproject.org/experiments/ENCSR668LDD/ | ENCODE, ENCFF616DLO |
| ENCODE Consortium | 2011 | K562 H3K9me3 ChIP | https://www.encodeproject.org/experiments/ENCSR000APE/ | ENCODE, ENCFF371GMJ |
| ENCODE Consortium | 2011 | K562 H3K4me1 ChIP | https://www.encodeproject.org/experiments/ENCSR000EWC/ | ENCODE, ENCFF159VKJ |
| Stergachis | 2020 | Fiber-seq of K562 cells | https://www.ncbi.nlm.nih.gov/geo/query/acc.cgi?acc=GSE146941 | NCBI Gene Expression Omnibus, SRP252718 |
| ENCODE Consortium | 2012 | K562 CTCF ChIP | https://www.encodeproject.org/experiments/ENCSR000EGM/ | ENCODE, ENCFF396BZQ |
| ENCODE Consortium | 2016 | K562 NRF1 ChIP | https://www.encodeproject.org/experiments/ENCSR837EYC/ | ENCODE, ENCFF626VDA |
| ENCODE Consortium | 2012 | K562 MYC ChIP | https://www.encodeproject.org/experiments/ENCSR000EGJ/ | ENCODE, ENCFF492XUU |
| ENCODE Consortium | 2011 | K562 PU.1 ChIP | https://www.encodeproject.org/experiments/ENCSR000BGW/ | ENCODE, ENCFF414ECK |
| ENCODE Consortium | 2011 | K562 GATA1 ChIP | https://www.encodeproject.org/experiments/ENCSR000EWM/ | ENCODE, ENCFF576YJD |
| ENCODE Consortium | 2014 | K562 REST ChIP | https://www.encodeproject.org/experiments/ENCSR137ZMQ/ | ENCODE, ENCFF290ESJ |

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
