## [Decision Letter]

**Acceptance summary:**

This manuscript describes a method, named SAMOSA, to identify nucleosome positions along chromatin segments that can be over 10 Kb in size. The approach employs EcoGII-modulated m^6^dA deposition on accessible non-nucleosomal DNA (inkers, nucleosome free regions) released from nuclear after mild MNase cleavage. The DNA modification is then read-out using PacBio sequencing. Mapping nucleosome positions along longer DNA stretches can provide information on variation in nucleosomal arrays, and how that relates to chromatin state and factor binding etc. The assay is validated using a reconstitute chromatin template and then applied to K562 cells, revealing significant variation in nucleosome positioning and nucleosome repeat lengths at transcription factor binding sites, and throughout domains with various histone modifications.

**Decision letter after peer review:**

Thank you for submitting your article "Massively multiplex single-molecule oligonucleosome footprinting" for consideration by *eLife*. Your article has been reviewed by three peer reviewers, including Job Dekker as the Reviewing Editor and Reviewer #1, and the evaluation has been overseen by Naama Barkai as the Senior Editor.

The reviewers have discussed the reviews with one another and the Reviewing Editor has drafted this decision to help you prepare a revised submission.

Summary:

This manuscript describes a method, named SAMOSA, to identify nucleosome positions along chromatin segments that can be over 10 Kb in size. The approach employs EcoGII-modulated m^6^dA deposition on accessible non-nucleosomal DNA (inkers, nucleosome free regions) released from nuclear after mild MNase cleavage. The DNA modification is then read-out using PacBio sequencing. Mapping nucleosome positions along longer DNA stretches can provide information on variation in nucleosomal arrays, and how that relates to chromatin state and factor binding etc. The assay is validated using a reconstitute chromatin template and then applied to K562 cells, revealing significant variation in nucleosome positioning and nucleosome repeat lengths at transcription factor binding sites, and throughout domains with various histone modifications.

Essential revisions:

Overall the approach works well and promises to address important questions, but the current work does not yet take full advantage of the single molecule nature of the assay and as such falls a bit short compared to very related methods that have recently been published (the works cited in the manuscript, and recently published work from the Stamatoyannopoulos lab). Can the authors acquire sufficient read coverage so that specific sites, e.g. specific CTCF sites are analyzed multiple times so that variation in the cell population at defined sites can be explored?

Many of the claims made about the potential of the method are insufficiently supported by the data provided. It appears that additional data is required to support the conclusions made from SAMOSA with respect to existing chromatin information, such as signal differences as a function of transcription factor binding (see below). The authors need to fully address these points in order for this manuscript to be considered for publication.

1) The authors should make an attempt to investigate where sequence bias influences a methylation call in their datasets. Clearly the pattern on the in vitro chromatinized template suggests that on average their methylated calls are correct. However, there appear to be clear positions in their chromatinized template datasets where this is not the case, i.e. lines in Figure 1—figure supplement 6A representing methylation calls in unmethylated template DNA and unmethylated calls on fully methylated template DNA. Upon close examination, this also seems the case in the chromatinized template, with certain positions inflexibly methylated/unmethylated and at odds with the surrounding linker/nucleosome patterning (Figure 1D). The authors should use K-mer analysis of methylated A's genome-wide to detect sequence bias in either the methyltransferase or sequencing platform.

2) It seems reasonable that the clustered data by NRL estimate (Figure 3) should correlate with existing measurements (i.e. MNase-seq). The authors should identify regions of the genome with strong enrichment for the seven clusters and compare this to nucleosome repeat length as can be estimated using conventional MNase measurements, i.e. the average distance between 5' mapping read positions across the genome (Valouev et al., 2011, Teif et al., 2012). Some agreement (for at least a few of these clusters with very regular nucleosomes) would strengthen the conclusions made by this approach, especially where there are irregular positioning patterns. Additionally, for these clusters the authors should display raw read alignment/methylation calls for SAMOSA at a few representative loci, where a sense of the raw data can be gleaned.

3) The comparisons of SAMOSA at different TF bound regions is likely influenced by the fraction of actually TF-bound molecules present in the original cellular sample. For example, CTCF is known to occupy it's strong motifs in the majority of cells, while few other factors have such regular binding/residency (Kelly et al., 2012 NomeSeq data at CTCF sites). It seems reasonable that some cluster fractions should scale with the enrichment for the factor (for at least CTCF and REST, the strong binding/nucleosome positioners), especially those associated with chromatin accessibility at the motif (i.e. A-accessible, HA-hyper-accessible). The authors should try to illustrate this, as well as representative read alignments/methylation calls at a few loci where these signals are prevalent.

4) The meta-plotted data seems noisy for most TFs profiled (Figure 4A-L) and the authors should show that their replicates agree with each other in terms of the relative size of clusters and at the metaplot level. Similarly, the data shown in Figure 5 should be broken into replicates. It is difficult to know to what extent the differences quoted are quantifiable/reproducible. For example, in panel A the reported deviation seems quite large around the median to make strong claims: e.g. "In specific cases, we observed small effect shifts in the estimated median NRLs for specific domains-for example, a shift of ~5 bp (180 bp vs. 185 bp) in H3K9me3 chromatin with respect to random molecules.…". This should also apply to the analysis done in Figure 5B and C, where it is difficult to get a sense of reproducibility from cluster size and the heatmap of Odds ratio and q-values.

The use of mild MNase is presented as an advantage, but is it really necessary? Adding EcoGII to isolated nuclei may work as well as shown in the recent Stamatoyannopoulos paper in Science.

In Figure 5, controls are randomly chosen nucleosomes, but it would be interesting to see what unmarked nucleosomes show. For example, unmarked alpha-satellite should be dominated by highly regular arrays with a 171-bp repeat length present in higher-order repeats corresponding to active centromeres, which consist of nucleosomal complexes that lack Histone H3 (CENP-A instead). The authors speculate that satellite irregularity might result from dynamic restructuring by HP1, and this predicts that other (H3-containing) unmarked satellites that lack H3K9me3 and presumably lack HP1 will be in regular arrays.

---

## [Author Response]

Essential revisions:Overall the approach works well and promises to address important questions, but the current work does not yet take full advantage of the single molecule nature of the assay and as such falls a bit short compared to very related methods that have recently been published (the works cited in the manuscript, and recently published work from the Stamatoyannopoulos lab). Can the authors acquire sufficient read coverage so that specific sites, e.g. specific CTCF sites are analyzed multiple times so that variation in the cell population at defined sites can be explored?

We are delighted that the reviewers agree on the importance of single-molecule chromatin profiling technologies like SAMOSA. In preparing this revision, we have addressed all of the major comments of reviewers, save one suggesting generating a high-coverage K562 dataset covering e.g. the same CTCF site multiple times (which we discuss in detail below). Our response broadly spans three areas: (1) providing dditional data while acknowledging existing challenges for sequencing individual sites to high coverage; (2) arguing how our manuscript uniquely leverages SAMOSA’s single-molecule resolution to uncover novel biological patterns, and (3) actively addressing all reviewer questions regarding the reproducibility of our primary findings.

Additional data and the challenge of high-coverage single-molecule sequencing. We completely agree that generating long-read datasets that demonstrate high (i.e. 50X – 1000X) coverage of individual TF binding sites is a desirable goal! However, we feel it important to highlight a key difference between SAMOSA and Illumina-based techniques that makes studying individual sites at high-coverage a challenge to be tackled in a separate study. Illumina sequencing methods are compatible with strategies like hybrid-capture or bisulfite PCR to deeply interrogate specific epigenomic regions^1,2^. Conversely, SAMOSA (and, indeed, the recently published Fiberseq work of Stergachis et al^3^) relies on native detection of modifications on unmodified and unamplified single templates. While this has significant benefits (i.e. non-destructive modification detection; access to long chromatin templates), it also makes deep sequencing of individual sites in a human-sized genome costly. Our eventual goal with SAMOSA is to generate high-coverage, haplotype-resolved maps, which will likely require additional methods development to selectively enrich for sites of interest. As such, we feel that deeply sequencing e.g. the K562 epigenome is beyond-scope for this proof-of-concept manuscript. In agreement with this point, we note that the Stergachis et al. paper does *not* provide a high-coverage K562 epigenome (reported coverage of 3.7X) – the majority of their study centers on the much smaller *D. melanogaster* genome (S2 cells). Still, to comply with the reviewers’ request for additional K562 SAMOSA data, we have sequenced additional experiments in the K562 cell line (new Figure 2—figure supplement 1). These shallowly-sequenced datasets further demonstrate the reproducibility of SAMOSA experiments (Figure 2—figure supplement 1B), as well as the ability to tune SAMOSA fragment length distributions by altering MNase digestion conditions (Figure 2—figure supplement 1A).

Leveraging single-molecule SAMOSA data using novel analytical strategies. We have taken steps to clarify how our study uniquely leverages the single-molecule nature of our assay. The majority of this paper inherently relies on SAMOSA’s single-molecule resolution, as we use the unbiased leiden community detection algorithm^4^ to define both genome-wide (from all sequenced molecules), and TF-centric clusters of single molecules on the basis of their respective m^6^dA methylation patterns. These analyses are only made possible by having access to individual sequenced templates, as opposed to population averages across genomic sites (i.e. Illumina data). Ultimately, we hope that the molecular and analytical methods outlined here provide a valuable jumping-off point for the broader application of long-read single-molecule footprinting to the study of gene regulation. In the revision, we specifically highlight this aspect of our study in the Discussion as follows:

“Our approach and associated analytical pipelines demonstrate the versatility of high-throughput single-molecule sequencing – namely the ability to cluster single-molecules in an unsupervised manner to uncover molecular states previously missed by short-read approaches. Our analytical approach bears many similarities to methods used in single-cell analysis, and indeed many of the technologies and concepts typically used for single-cell genomics^5^ (e.g. clustering; trajectory analysis) will likely have value when applied to single-molecule epigenomic assays.”

Addressing questions of reproducibility across SAMOSA replicates, and validating the primary biological conclusions of our study. We recognize the need to more concretely demonstrate the reproducibility of our SAMOSA experiments and experimental findings. To this end, we have followed all of the remaining reviewers’ suggestions, including (i) adding supplementary figures faceted by biological replicates, (ii) validating the primary finding of our paper (heterogeneity of chromatin fibre structures, and enrichment for irregular fiber composition in constitutive heterochromatin) using data from Stergachis et al., and finally (iii) we have provided additional light sequencing data of a set of K562 SAMOSA libraries to demonstrate that SAMOSA signals are en masse reproducible, and can be used to profile longer chromatin fibres by tuning MNase digest conditions. Moreover, we have ensured that our revised manuscript has no broad biological claims beyond those firmly supported by our replicated analyses – namely, the finding that chromatin domains are comprised of many different fibre types, and that specific irregular or regular fibre types are differentially employed by specific chromatin domains.

Many of the claims made about the potential of the method are insufficiently supported by the data provided. It appears that additional data is required to support the conclusions made from SAMOSA with respect to existing chromatin information, such as signal differences as a function of transcription factor binding (see below). The authors need to fully address these points in order for this manuscript to be considered for publication.

In our primary submission, we strove to present proof-of-concept of the SAMOSA method while avoiding broad biological claims unfounded by our data. As you will see below, we have addressed the primary issues raised by reviewers, which largely pertain to the reproducibility of our replicates and major findings. Moreover, we have softened the tone of the manuscript in cases where reviewers disagreed with claims as stated. For example, we have removed text and figures concerning the coverage of SAMOSA libraries versus other approaches, as suggested.

1) The authors should make an attempt to investigate where sequence bias influences a methylation call in their datasets. Clearly the pattern on the in vitro chromatinized template suggests that on average their methylated calls are correct. However, there appear to be clear positions in their chromatinized template datasets where this is not the case, i.e. lines in Figure 1—figure supplement 6A representing methylation calls in unmethylated template DNA and unmethylated calls on fully methylated template DNA. Upon close examination, this also seems the case in the chromatinized template, with certain positions inflexibly methylated/unmethylated and at odds with the surrounding linker/nucleosome patterning (Figure 1D). The authors should use K-mer analysis of methylated A's genome-wide to detect sequence bias in either the methyltransferase or sequencing platform.

This is a fantastic suggestion, and one that we have now included in our revised manuscript. To address this question, we have analyzed positive control methylation samples and negative controls from our array experiments and quantified interpulse duration (IPD) differences observed as a function of different *k*-mers (Figure 1—figure supplement 4). Specifically, we examined distributions of normalized IPD values (before our methylation calling) of represented 10-mer sequences in both the negative and positive control. We binned the adenines in our template DNA by sequence context from 2 basepairs (bp) upstream to 7 bp downstream of the template adenine, the context previously found to significantly determine IPD values^6^. We then visualized the average IPD of individual 10-mer sequences as a heatmap for negative, combined, and positive controls. Briefly, we find that a portion of sequence contexts show incomplete methylation in our positive control, and that GATC motifs are constitutively methylated in unmethylated controls. Given that array DNA was propagated in *Dam*^+^ bacteria (necessary due to synthetic lethality of *Dam*^-^/*RecA*^-^*E. coli*), this finding is completely expected (and explains the ‘stripes’ seen in new Figure 1—figure supplement 6). These contexts also appear to reveal a slight sequence specificity of EcoGII, which appears to methylate A/T-rich contexts slightly less efficiently. We have added this figure to the manuscript and added associated text explaining these biases:

“These patterns were subtly influenced by the associated 10-mer context of sequenced bases, consistent with possible enzymatic biases, but also previous observations of sequence-influenced shifts in polymerase kinetics (Figure 1—figure supplement 4).”

Explicitly modeling the native kinetic differences by which the PacBio polymerase navigates modified and unmodified *k*-mers is a significant computational undertaking by itself; we hope that this type of *k*-mer stratification will ultimately enable more sophisticated analysis of relationships between sequence context and interpulse duration on the PacBio platform.

2) It seems reasonable that the clustered data by NRL estimate (Figure 3) should correlate with existing measurements (i.e. MNase-seq). The authors should identify regions of the genome with strong enrichment for the seven clusters and compare this to nucleosome repeat length as can be estimated using conventional MNase measurements, i.e. the average distance between 5' mapping read positions across the genome (Valouev et al., 2011, Teif et al., 2012). Some agreement (for at least a few of these clusters with very regular nucleosomes) would strengthen the conclusions made by this approach, especially where there are irregular positioning patterns. Additionally, for these clusters the authors should display raw read alignment/methylation calls for SAMOSA at a few representative loci, where a sense of the raw data can be gleaned.

A major advantage of SAMOSA and similar approaches is that they capture nucleosomal patterning that is inherently obfuscated by mapping MNase fragment-ends or midpoints (due to both inherent fibre irregularity, or lack of consistent phase of regular fibres across many haplotypes). Still, we appreciate the need to validate the major conclusion of our proof-of-concept study, which is that constitutive heterochromatic domains harbor a mixture of short NRL fibres and irregular chromatin fibres, and that e.g. Polycomb-repressed domains are enriched for the longest NRL fibres. To do this, we have assessed the reproducibility of our principle biological claims in two ways. First, we have stratified our data by replicate, and can show that the relative cluster enrichment / depletion we see for each chromatin state is highly reproducible between replicates (new Figure 6—figure supplement 1). Second, we have sought out an orthogonal way to validate these findings, by reprocessing the recently published data of Stergachis et al., who use a completely different enzyme and set of reaction conditions to footprint K562 chromatin (new Figure 6—figure supplement 2). We reprocessed data from the Stergachis et al. K562 dataset and clustered 2.5E5 molecules using our analytical pipeline. We find that molecules fall into five broad clusters (Figure 6—figure supplement 2A)—one irregular cluster (annotated as IR_S_ for Irregular-Stergachis *et al*), and four regular clusters sorted by increasing estimated NRL. We then estimated the relative usage of these clusters across epigenomic domains as in our initial manuscript (Figure 6—figure supplement 2B). Consistent with our initial findings, we find that H3K9me3 decorated chromatin is enriched for fibre-type IR_S_ and the shortest NRL class of fibres. Additionally, we also find that this reanalysis confirms our finding that H3K27me3 domains harbor the longest NRLs. Taken together, this reanalysis serves as a powerful confirmation of our preliminary findings. Finally, given the low per-site coverage of our data, we believe that the single-molecule representations shown in our submission (i.e. individual molecules and associated signal; for example Figure 4M) are the most honest representation of our raw data—the alternative would be showing between 1 – 3 molecules covering a given site, which is such a low number of observations that readers may wrongly ascribe biological relevance to observed methylation patterns.

3) The comparisons of SAMOSA at different TF bound regions is likely influenced by the fraction of actually TF-bound molecules present in the original cellular sample. For example, CTCF is known to occupy it's strong motifs in the majority of cells, while few other factors have such regular binding/residency (Kelly et al., 2012 NomeSeq data at CTCF sites). It seems reasonable that some cluster fractions should scale with the enrichment for the factor (for at least CTCF and REST, the strong binding/nucleosome positioners), especially those associated with chromatin accessibility at the motif (i.e. A-accessible, HA-hyper-accessible). The authors should try to illustrate this, as well as representative read alignments/methylation calls at a few loci where these signals are prevalent.

We agree that this is an interesting analysis, and have promoted this panel to the main text (new Figure 5A). To the point of examining the relative enrichment / depletion of individual sites, we note that this analysis requires very high coverage of individual predicted binding sites. As generating high coverage of individual CTCF sites is beyond scope for this study, we did not include such an analysis in our original submission or revision but we agree that it will be interesting to pursue after e.g. target enrichment methods are well-optimized for unamplified PacBio sequencing libraries. Moreover, we have (as discussed below), carried out reproducibility analyses across our replicates to demonstrate that the clusters shown in Figure 4 and their respective enrichment across different TFs surveyed are reproducible (new Figure 5—figure supplement 1). We have tried to address this honestly in our manuscript in the following clause:

“While correlation of our replicates demonstrates the reproducibility and robustness of these findings (Figure 5—figure supplement 1), future experimental follow-up coupling our protocol with perturbed biological systems and deeper sequencing are necessary to quantitatively interrogate this model.”

4) The meta-plotted data seems noisy for most TFs profiled (Figure 4A-L) and the authors should show that their replicates agree with each other in terms of the relative size of clusters and at the metaplot level. Similarly, the data shown in Figure 5 should be broken into replicates. It is difficult to know to what extent the differences quoted are quantifiable/reproducible. For example, in panel A the reported deviation seems quite large around the median to make strong claims: e.g. "In specific cases, we observed small effect shifts in the estimated median NRLs for specific domains-for example, a shift of ~5 bp (180 bp vs. 185 bp) in H3K9me3 chromatin with respect to random molecules.…". This should also apply to the analysis done in Figure 5B and C, where it is difficult to get a sense of reproducibility from cluster size and the heatmap of Odds ratio and q-values.

We thank the reviewer for bringing this point to our attention. We have repeated this analysis stratified by replicate and included these as Figure 5—figure supplement 1. This analysis firmly shows that our observed cluster sizes, patterns, and enrichment / depletion across the different TF binding sites surveyed here are reproducible. Regarding our description of the NRL deviations observed in our data, we agree that these deviations are quite large, and hence noted that any observed shifts are very small effects (i.e. not a ‘strong claim’ of effect). Regardless, the reproducibility of our NRL class enrichments across replicates (Figure 6—figure supplement 1), and the replication of these trends in a completely different dataset (Figure 6—figure supplement 2) sufficiently speak to the robustness of the biological claims made in this proof-of-concept paper.

The use of mild MNase is presented as an advantage, but is it really necessary? Adding EcoGII to isolated nuclei may work as well as shown in the recent Stamatoyannopoulos paper in Science.

As our reanalysis of Stergachis et al’s K562 data demonstrates, the reaction conditions for Fiberseq appear to enable footprinting of fibers in constitutive heterochromatin; modifying our existing SAMOSA workflow to include m^6^dA methylation before MNase digestion is an exciting idea that could be pursued in later studies. We note this in the revision as follows:

“Second, our approach involves methylating fibres following solubilization of oligonucleosomal fragments, and is thus unlikely to capture protein-DNA interactions weaker or more transient than the stable nucleosome-DNA interaction. Such transient interactions could be captured in future work by modifying the protocol to footprint nuclei prior to MNase-solubilization.”

At the same time, the use of mild MNase and solubilization *is* an advantage because it provides two orthogonal measures of chromatin structure. First, our molecules have 2 MNase-cleavages that footprint protein-DNA interactions (as shown in Figure 4A-G). Second, our footprinted fibres have m^6^dA signal that provides single-molecule nucleosome positioning information. In this way, our assay combines information captured by an assay like Array-seq^7^ with methylation footprinting. In our revision, we also demonstrate that the MNase digestion in the SAMOSA protocol is tunable: we provide SAMOSA data for three additional digestion conditions, resulting in different fragment length distributions, while still maintaining the classical periodic patterning of m^6^dA modification signal seen in previous SAMOSA experiments (shown in new Figure 2—figure supplement 1).

In Figure 5, controls are randomly chosen nucleosomes, but it would be interesting to see what unmarked nucleosomes show. For example, unmarked alpha-satellite should be dominated by highly regular arrays with a 171-bp repeat length present in higher-order repeats corresponding to active centromeres, which consist of nucleosomal complexes that lack Histone H3 (CENP-A instead). The authors speculate that satellite irregularity might result from dynamic restructuring by HP1, and this predicts that other (H3-containing) unmarked satellites that lack H3K9me3 and presumably lack HP1 will be in regular arrays.

Many thanks to the reviewer for this insightful suggestion! To address this, we have stratified the ‘Satellite’ label into alpha-, beta-, and gamma-satellite sequences, and repeated our enrichment analysis across the 7 fibre types originally described in our manuscript. While alpha and beta satellite sequences are enriched for the irregular fibre type (which we speculate may in part be mediated by HP1-driven chromatin deformation), gamma satellite sequence is instead depleted for irregular fibres and enriched for long NRL regular fibres. Consistent with the reviewers’ astute observations, and the work of Kim et al., gamma-satellite is *expected* to be unmarked by H3K9me3^8^. Given that this strengthens our prior speculation that the irregular fibres we are observing could be due to HP1-mediated deformation of heterochromatin, we have added this analysis (new Figure 6—figure supplement 3) and associated text to the revision, as replicated below:

“Given the robustness of this finding, it is tempting to speculate that this irregularity may be linked to the dynamic restructuring of heterochromatic nucleosomes by factors like HP1^9^, which may promote phase-separation of heterochromatin. While stratification of analyzed satellite sequences into H3K9me3-decorated alpha / beta, and H3K9me3-free gamma satellite provides correlative support for this notion (Figure 6—figure supplement 3), future studies combining SAMOSA with cellular perturbation of heterochromatin-associated factors are necessary to directly address this possibility.”

References:

1) Krebs, A. R. et al. Genome-wide Single-Molecule Footprinting Reveals High RNA Polymerase II Turnover at Paused Promoters. Molecular Cell 67, 411–422.e4 (2017).

2) Sönmezer, C. et al. Single molecule occupancy patterns of transcription factors reveal determinants of cooperative binding in vivo. bioRxiv 2020.06.29.167155 (2020).

3) Stergachis, A. B., Debo, B. M., Haugen, E., Churchman, L. S. and Stamatoyannopoulos, J. A. Single-molecule regulatory architectures captured by chromatin fiber sequencing. Science 368, 1449–1454 (2020).

4) Traag, V. A., Waltman, L. and van Eck, N. J. From Louvain to Leiden: guaranteeing wellconnected communities. Scientific reports 9, 1–12 (2019).

5) Trapnell, C. Defining cell types and states with single-cell genomics. Genome Research 25, 1491–1498 (2015).

6) Feng, Z. et al. Detecting DNA modifications from SMRT sequencing data by modeling sequence context dependence of polymerase kinetic. PLoS Comput Biol 9, e1002935 (2013).

7) Baldi, S., Krebs, S., Blum, H. and Becker, P. B. Genome-wide measurement of local nucleosome array regularity and spacing by nanopore sequencing. Nat Struct Mol Biol 25, 894–901 (2018).

8) Kim, J.-H. et al. Human γ-satellite DNA maintains open chromatin structure and protects a transgene from epigenetic silencing. Genome Research 19, 533–544 (2009).

9) Sanulli, S. et al. HP1 reshapes nucleosome core to promote phase separation of heterochromatin. Nature 575, 390–394 (2019).